**Impacts of Three Gorges Dam's operation on spatial-temporal**
**patterns of tide-river dynamics in the Yangtze River estuary, China**
Huayang Cai[1, 2, 3, 4], Xianyi Zhang[1, 2, 3], Leicheng Guo [4], Min Zhang[5,*], Feng Liu[1, 2, 3],
Qingshu Yang[1, 2, 3]
*1. Institute of Estuarine and Coastal Research, School of Marine Engineering and*
*Technology, Sun Yat-sen University, Guangzhou, China*
*2. Guangdong Provincial Engineering Research Center of Coasts, Islands and Reefs,*
*Guangzhou, China*
*3. Southern Marine Science and Engineering Guangdong Laboratory (Zhuhai),*
*Zhuhai, China*
*4. State Key Laboratory of Estuarine and Coastal Research, East China Normal*
*University, Shanghai, China*
*5. Shanghai Normal University, School of Environmental and Geographical Sciences,*
*Shanghai, China*
**Corresponding author:** Min Zhang
**Corresponding author's E-mail:** zhangmin@shnu.edu.cn
**Key points**
1.  Impacts of TGD operation on tide-river dynamics are quantified using an analytical

21       model.

2.  The strongest impacts occurred during autumn and winter due to the seasonal

23       freshwater regulation by TGD.

3.  The alteration of tide-river dynamics may exert considerable impacts on sustainable

25       water resource management in dam-controlled estuaries.

**Abstract**

The Three Gorges Dam (TGD), located in the mainstream of the Yangtze River, is the world's largest hydroelectric station in terms of installed power capacity. It was demonstrated that the TGD had caused considerable modifications in the downstream freshwater discharge due to its seasonal operation mode of multiple utilisation for flood control, irrigation, and power generation. To understand the impacts of the freshwater regulation of TGD, an analytical model is adopted to explore how the operation of TGD may affect the spatial-temporal patterns of tide-river dynamics in the Yangtze River estuary. We evaluated the effect of TGD by comparing the changes in major tide-river dynamics in the post-TGD period (2003–2014) with those in the pre-TGD period (1979–1984). The results indicate that the strongest impacts occurred during the autumn and winter, corresponding to a substantial reduction in freshwater discharge during the wet-to-dry transition period and slightly increased discharge during the dry season. The underlying mechanism leading to changes in the tide-river dynamics lies in the alteration of freshwater discharge, while the impact of geometric change is minimal. Overall, the results suggest that the spatial-temporal pattern of tide-river dynamics is sensible to the freshwater regulation of the TGD, so that the ecosystem function of the estuary may undergo profound disturbances. The results obtained from this study can be used to set scientific guidelines for water resource management (e.g. navigation, flood control, salt intrusion) in dam-controlled estuarine systems.

**Key words:** seasonal freshwater regulation, Three Gorges Dam, analytical model, tide-river dynamics, Yangtze River estuary

## 1. Introduction

Estuaries are transition zones where river meets ocean (Savenije, 2012). Tide-river interactions, a result of both hydrologic drivers and geomorphic constraints, are highly dynamic in estuaries (Buschman et al., 2009; Sassi and Hoitink, 2013; Guo et al., 2015; Cai et al., 2016; Hoitink and Jay, 2016; Hoitink et al., 2017; Du et al., 2018). In natural conditions, they usually experience a wide range of temporal variations, in timescale ranging from a fortnight to season (e.g. Zhang et al., 2018). Human intervention, such as dam construction in the upstream parts of a river and the growing number of water conservancy projects built along large rivers (such as freshwater withdrawal), have caused seasonal changes in downstream freshwater discharge delivery, leading to adjustments in the function of fluvial and estuarine hydrology (e.g. Lu et al., 2011; Mei et al., 2015; Dai et al., 2017). Consequently, it is important to understand the impacts of large-scale human intervention, such as flood control, navigation, salt intrusion, and freshwater withdrawal, which are relevant not only to tide-river dynamics and riparian ecology but also to sustainable water resource management in general.

River discharge generally fluctuates following a wet-dry cycle due to the seasonal variation of precipitation in the upstream river basin. For instance, the Yangtze River, the largest river in China in terms of mean discharge, which flows into the East China Sea, has a maximum river discharge during summer in July and a low value during winter in January, with a maximum discharge difference of approximately 38,000 $m^3$/s (Cai et al., 2016). Similar seasonal variations are also identified in other large rivers in

eastern and southern Asia, such as the Mekong River in Vietnam, Ganges River in India,
and Pearl River in China, under the influence of a monsoon climate. However, most
large rivers have been significantly dammed at the central and upper reaches in recent
decades, dramatically modifying stream hydrology and sediment delivery, resulting in
changes in hydraulics and river delta development trend at the lower reaches (e.g.
Räsänen et al, 2017; Rahman et al., 2018; Liu et al., 2018). Due to the fact that the
response of tide-river interactions to the impacts of dams are diverse and non-uniform
and that many more dams are to be built in the future, the impacts of the hydrodynamic
interactions between tidal waves and seasonal river flows from natural variations and
anthropogenic activities have become a common focus in international hydraulic
research, especially in large tidal rivers.

The Yangtze River estuary, located near the coastal area of East China Sea, is one of the
largest estuaries in Asia. In the mouth of the Yangtze River estuary, bifurcation occurs
and the characteristics of tides have been broadly investigated in previous studies (e.g.
Zhang et al., 2012; Lu et al., 2015; Alebregtse and Swart, 2016). However, in these
studies, river influences are usually neglected. In recent years, the processes of
nonlinear interactions between tidal wave and river flow in the Yangtze River estuary
have received increasing attention (e.g. Guo et al., 2015; Zhang et al., 2015a, b; Cai et
al., 2016; Kuang et al., 2017; Zhang et al., 2018). However, recent studies on tidal
properties, such as asymmetry, changes near the mouth area, and seasonal variations in
tidal wave propagation and fluvial effects over the entire 600 km of the tidal river, up
to the tidal limit of the Datong hydrological station, have been limited. In addition, the
operation of the Three Gorges Dam (TGD), the largest dam in the world, has
substantially affected the downstream river hydrology and sediment delivery. There is
a variety of debate regarding the potential impacts of TGD on the downstream river
morphology, hydrology, and ecology, since the underlying mechanism of the impact of
the TGD is not fully understood. Specifically, the TGD operation has altered the
downstream fluvial discharge and water levels on the seasonal scale, directly following
the reservoir seasonal impounding and release of water volume (e.g. Chen et al., 2016;
Guo et al., 2018). However, the impacts of seasonal freshwater regulation by the TGD
on the spatial-temporal tide-river dynamics in the downstream estuarine area have not
been systematically investigated. For example, during the dry season TGD operation
increased the multi-year monthly averaged river discharge at Datong station from 9520
$m^3 \cdot s^{-1}$ to 12896 $m^3 \cdot s^{-1}$ in January, while during wet season the regulation reduced the
river discharge from 49900 $m^3 \cdot s^{-1}$ to 44367 $m^3 \cdot s^{-1}$ in July during the pre- and post- TGD
period.

In this study, for the first time, the spatial-temporal variations in the hydrodynamic
processes due to the interactions of tidal flow and fluvial discharge in the Yangtze River
estuary caused by natural forcing and human intervention were studied, with specific
focus on the effect of TGD seasonal regulation. Here, we adopted a well-developed
analytical model proposed by Cai et al. (2014a, 2016) to investigate the spatial-temporal
patterns of tide-river dynamics in the entire Yangtze River estuary and quantify the
impacts of the TGD operation. In the following sections, we introduce the study site of
the Yangtze River estuary. This is followed by a description of the available data and
analytical model of tide-river dynamics in Section 3. Subsequently, we applied the
model to the Yangtze River estuary, where the TGD has operated since 2003 (Section
4). In particular, we explored the alteration of the tide-river dynamics after the TGD
closure and summarise the impacts of the TGD on the spatial-temporal patterns of tide-
river dynamics. The impacts of channel geometry and river discharge alterations on
tide-river dynamics as well as the implications for sustainable water resource
management were then discussed in Section 5. Finally, some key findings were
addressed in Section 6.

**2.  Overview of the Yangtze River estuary**
The Yangtze River, flowing from west to east in central China, is one of the world's
most important rivers due to its great economic and social relevance. It has a length of
about 6300 km and a basin area of about 190,000 km$^2$ (Figure 1a). The Yangtze River
basin is geographically divided into three parts, the upper, central, and lower sub-basins,
and contains an estuary area with partitions at Yichang, Jiujiang, and Datong (DT),
respectively (Figure 1a). Of concern in this study are the impacts of the Three Gorges
Dam (TGD), the world's largest dam, on the spatial-temporal patterns of tide-river
dynamics in the estuarine area. It is located about 45 km upstream of Yichang (Figure
1a). The TGD project began in 2003; by 2009, when full operations began, the total
water storage capacity rose up to ~40 km$^3$, equivalent to 5% of the Yangtze's annual
discharge. Downstream the DT station, where the tidal limit is located, the Yangtze
River estuary extends ~630 km to the seaward end of the South Branch. Wuhu (WH),
Maanshan (MAS), Nanjing (NJ), Zhenjiang (ZJ), Jiangyin (JY), and Tianshenggang
(TSG) are the major gauging stations along the mainstream in the seaward direction
(Figure 1b). Under the control of the Asian monsoon climate, river discharges show
distinct seasonal patterns. In 1979–2012, more than 70% of freshwater was discharged
at DT occurred during summer (May–October).

Apart from river flows, tidal waves are also recognised as the major sources of energy
for hydrodynamics in the Yangtze River estuary, which is characterised by a meso-tide
with a tidal range of up to 4.6 m and a mean tidal range of ~2.7 m near the estuary
mouth. According to the observation in the Gaoqiaoju tidal gauging station (1950–
2012), the averaged ebb tide duration (7.5 h) is a bit longer than the averaged flood tide
duration (5 h), indicating an irregular semidiurnal character (Zhang et al., 2012).

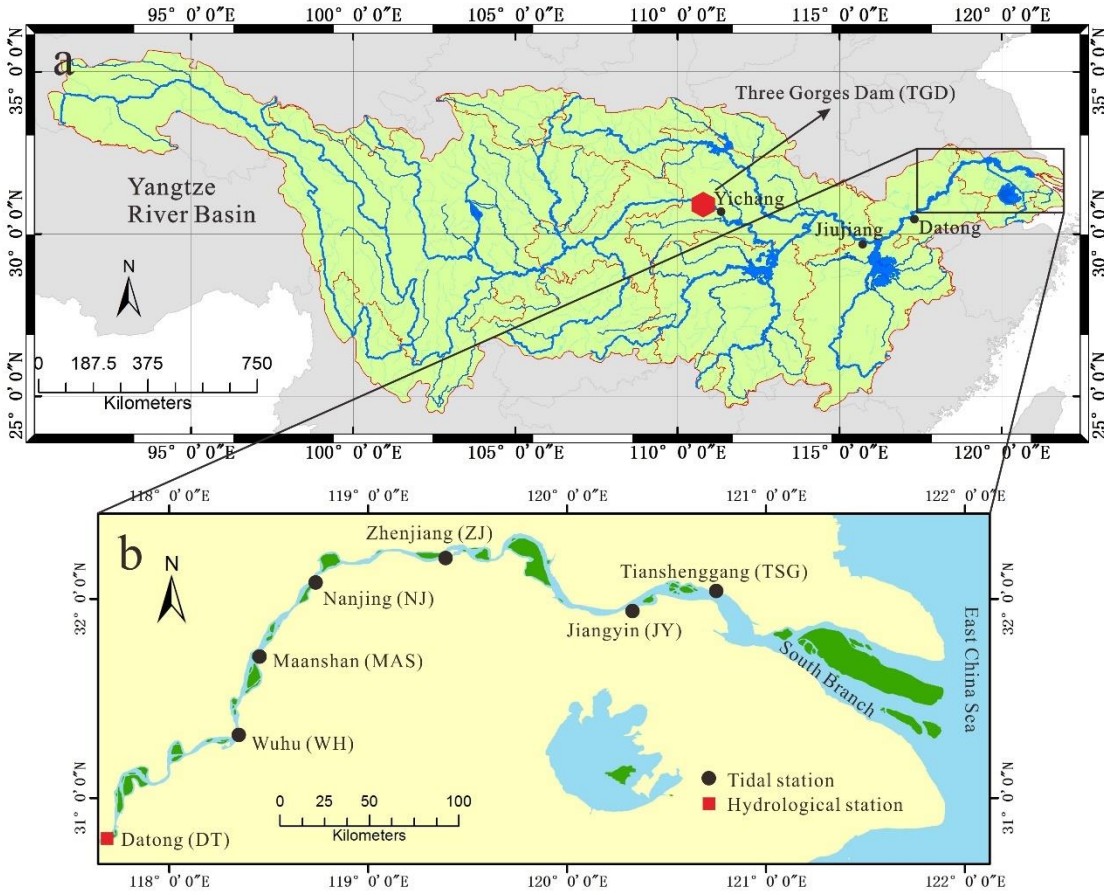


Figure 1. Maps of the Yangtze River basin (a) and Yangtze River estuary (b) with the
location of tidal gauging and hydrological stations shown with black solid circles and

157                              red solid rectangles.


## 3.  Data and Methodology

### 3.1 Source of Data

To quantitatively investigate the relationship between freshwater discharge regulation

caused by the TGD operation and the tide-river dynamics, monthly averaged

hydrological data for both pre-TGD (1979–1984) and post-TGD (2003–2014) periods

of tidal range and water level from the above-mentioned six tidal gauging stations along

the Yangtze River estuary were collected. They were published by the Yangtze

Hydrology Bureau of the People's Republic of China. The monthly averaged tidal
amplitude is determined by averaging the daily difference between high and low water
levels and dividing by two. To correctly quantify the residual water level along the
Yangtze estuary, locally measured water level at different gauging stations are corrected
to the national mean sea level of Huanghai 1985.

**3.2 Analytical model for tide-river dynamics**
**3.2.1 Basic equations**
In tidal rivers, the tidally averaged water level (i.e. residual water level) depicts a steady
gradient, which usually increases with freshwater discharge (e.g. Sassi and Hoitink,
2013). The key to deriving the dynamics of the residual water level lies in the one-
dimensional momentum equation, which can be expressed as (e.g. Savenije, 2005,

178 2012):

$$\frac{\partial U}{\partial t} + U \frac{\partial U}{\partial x} + g \frac{\partial Z}{\partial x} + \frac{gh}{2\rho} \frac{\partial \rho}{\partial x} + g \frac{U|U|}{K^2 h^{4/3}} = 0, \qquad (1)$$

where $U$ is the cross-sectional averaged velocity, $Z$ is the free surface elevation, $h$ is the
water depth, $g$ is the acceleration due to gravity, $t$ is the time, $\rho$ is the water density, $x$ is
the longitudinal coordinate directed landward, and $K$ is the Manning-Strickler friction
coefficient. It was demonstrated that in the subtidal momentum balance, the residual
water level slope is primarily balanced by the residual friction term (Vignoli et al., 2003;
Buschman et al., 2009; Cai et al., 2014a, for a detailed derivation, readers can refer to
the Appendix A):
$$\overline{\frac{\partial Z}{\partial x}} = -\overline{\frac{U|U|}{K^2 h^{4/3}}} \qquad (2)$$

where the overbars indicate the tidal average. For a single channel with the residual water level set to 0 at the estuary mouth (i.e. $\bar{Z} = 0$ at $x = 0$), the integration of Equation (2) leads to an analytical expression for the residual water level

$$\bar{Z}(x) = -\int_0^x \overline{\frac{\partial Z}{\partial x}} = -\int_0^x \overline{\frac{U|U|}{K^2 h^{4/3}}} \quad .\tag{3}$$

To derive the analytical solutions for tide-river dynamics, we assume that the longitudinal variation of cross-sectional area $\bar{A}$ and width $\bar{B}$ can be described by the following exponential functions (see also Toffolon et al., 2006; Cai et al., 2014a):

$$\bar{A} = \overline{A_r} + \left(\overline{A_0} - \overline{A_r}\right)\exp\left(-\frac{x}{a}\right),\tag{4}$$

$$\bar{B} = \overline{B_r} + \left(\overline{B_0} - \overline{B_r}\right)\exp\left(-\frac{x}{b}\right),\tag{5}$$

where $\overline{A_0}$ and $\overline{B_0}$ represent the tidally averaged cross-sectional area and width at the estuary mouth, respectively, $\overline{A_r}$ and $\overline{B_r}$ represent the asymptotic riverine cross-sectional area and width, respectively, and $a$ and $b$ are the convergence lengths of the cross-sectional area and width, respectively. The advantage of these equations for approximating the shape of the estuary is that they account not only for the exponential shape in the lower part of the tidal river but also for the approximately prismatic channel in the upstream part of the tidal river. We further assume a nearly rectangular cross-section, considering a large width to depth ratio; hence, the tidally averaged depth is given by $\bar{h} = \bar{A}/\bar{B}$ and the cross-sectional area variability can be primarily attributed to the change in depth.

**3.2.2  Analytical solution for tidal hydrodynamics**

It was shown by Cai et al. (2014a, b, 2016) that the tide-river dynamics is dominantly
controlled by four dimensionless parameters (see their definitions in Table 1). They
include: the dimensionless tidal amplitude $\zeta$ (representing the boundary condition in the
seaward side), the estuary shape number $\gamma$ (representing the cross-sectional area
convergence), the friction number $\chi$ (representing the bottom frictional effect), and the
dimensionless river discharge $\varphi$ (representing the impact of freshwater discharge). The
definitions of these four variables are defined in Table 1, where $\eta$ is the tidal amplitude,
$\upsilon$ is the velocity amplitude, $U_r$ is the river flow velocity, $\omega$ is the tidal frequency, $r_S =$
$B_S/\bar{B}$ is the storage width ratio between the storage width $B_S$ and the stream width $\bar{B}$
that accounts for the effect of storage area (i.e., tidal flats or salt marshes), and $c_0$ is the
classical wave celerity defined as $c_0 = \sqrt{g\bar{h}/r_S}$ .
Table 1. Definitions of dimensionless parameters used in the analytical model

| Local variables | Dependent variables |
| --- | --- |
| Dimensionless tidal amplitude | Amplification number |
| $\zeta = \eta / \bar{h}$ | $\delta = c_0 \, \mathrm{d}\eta / (\eta \omega \mathrm{d}x)$ |
| Estuary shape number | Velocity number |
| $\gamma = c_0\left(\bar{A} - \bar{A}_r\right)/\left(\omega a \bar{A}\right)$ | $\mu = \upsilon / (r_s \zeta c_0) = \upsilon \bar{h} / (r_s \eta c_0)$ |
| Friction number | Celerity number |
| $\chi = r_S g c_0 \zeta \left[1 - (4\zeta/3)^2\right]^{-1} / \left(\omega K^2 \bar{h}^{4/3}\right)$ | $\lambda = c_0 / c$ |
| Dimensionless river discharge | Phase lag |
| $\varphi = U_r / \upsilon$ | $\varepsilon = \pi/2 - (\phi_Z - \phi_U)$ |


In this study, we used the analytical solutions proposed by Cai et al. (2014a, b, 2016),
in which the solutions of the major tide-river dynamics are derived by solving a set of
four implicit equations for the tidal damping, the velocity amplitude, the wave celerity,
and the phase lag (see details in Appendix B). The major dependent parameters can be
described by the following four variables (see also Table 1): $\delta$ represents the
damping/amplification number describing the increase ($\delta > 0$), or decrease ($\delta < 0$) of
the tidal wave amplitude along the estuary axis, $\mu$ represents the velocity number
indicating the ratio of actual velocity amplitude to the frictionless value in a prismatic
channel, $\lambda$ represents the celerity number representing the classical wave celerity $c_0$
scaled by the actual wave celerity $c$, and $\varepsilon$ represents the phase lag between the high
water (HW) and high water slack (HWS) or between the low water (LW) and low water
slack (LWS). It is important to note that the phase lag (ranging between 0 and $\pi/2$) is a
key parameter in classifying the estuary, where $\varepsilon = 0$ suggests the tidal wave is featured
by a standing wave, while $\varepsilon = \pi/2$ indicates a progressive wave. For a simple harmonic
wave, the phase lag is defined as $\varepsilon = \pi/2 - (\phi_Z - \phi_U)$, where $\phi_Z$ and $\phi_U$ are the phases
of elevation and current, respectively (Savenije, et al., 2008).

**3.2.3 Analytical solution for the entire channel**
It is worth noting that the analytically computed tide-river dynamics $\mu$, $\delta$, $\lambda$, and $\varepsilon$ only
represent local hydrodynamics since they depend on local (fixed position) values of the
dimensionless parameters, i.e. the tidal amplitude $\zeta$, the estuary shape number $\gamma$, the
friction number $\chi$, and the river discharge $\varphi$ (see Table 1). To correctly reproduce the
tide-river dynamics for the entire channel, a multi-reach technique is adopted by
subdividing the entire estuary into multiple reaches to account for the longitudinal
variations of the estuarine sections (e.g. bed elevation, bottom friction). For a given
tidal damping/amplification number $\delta$ and tidal amplitude $\eta$ at the seaward boundary, it
is possible to determine the tidal amplitude at a distance $\Delta x$ (e.g. 1 km) upstream by
simple explicit integration. Hence, the analytical solution for the entire channel can be
obtained by step-wise integration in this way.

**4. Results**
**4.1 Observational analysis on the alteration of tide-river dynamics after TGD**
**closure**
To quantify the impacts of TGD operation on the downstream tide-river dynamics, we
divided the time series into two periods, including a pre-TGD period (1979–1984,
representing the condition before the operation of the TGD) and a post-TGD period
(2003–2014, after the closure of the TGD with an operating TGD). Figure 2 shows the
changes in the observed tidal range $\Delta H$ and residual water level $\Delta \bar{Z}$ before and after
the closure of the TGD at the six gauging stations, together with the change in
freshwater discharge $\Delta Q$ observed at the DT hydrological station. Figure 2 and Table 2
clearly show that the monthly averaged river discharge in January, February, and March
substantially increased by 35.5%, 30.5%, and 16.4%, respectively, due to the
considerable release of freshwater from the TGD. On the other hand, we observe a
significant decrease in freshwater discharge in September, October, and November,

decreasing by 20.1%, 33.2%, and 20.8%, respectively. The reason can be primarily attributed to the impounding water of the TGD during these months, especially in October. During the other months, the impacts of TGD on the change in the freshwater discharge are relatively small, mimicking the natural condition before the operation of the TGD.

In Figure 2a we observe an increasing trend in tidal range for the post-TGD period at the six gauging stations, except for the marked decrease at the ZJ station in the first half of the year (i.e. January–June). On average, the maximum increase (0.20 m) in tidal range occurs in October, which is mainly due to the substantial reduction of river discharge caused by the TGD operation. This indicates a consistent enhancement of tidal dynamics along the Yangtze estuary, except the reach near the ZJ station. The exceptional case in ZJ station is likely due to the fact that ZJ station is located near the position of the tidal current limit during the dry season (Guo et al., 2015; Zhang et al., 2018). The shallow and narrow geometry around ZJ station impedes the tidal wave propagation when river discharge increases due to the TGD operation during the dry season (Chen et al., 2012), leading to a remarkably decreasing tidal range in the first half of the year. For the residual water level, Figure 2b clearly shows that the change in the residual water level directly follows that of the river discharge due to the stable relationship between these two parameters. In particular, we see that the residual water levels increased by 0.26 m, 0.30 m, and 0.16 m, respectively, in January, February, and March, while they significantly decreased by 0.72 m, 1.17 m, and 0.70 m, respectively,

in September, October, and November. In addition, the decrease trend in residual water

level is more significant at upstream stations when compared with those in the

downstream areas.

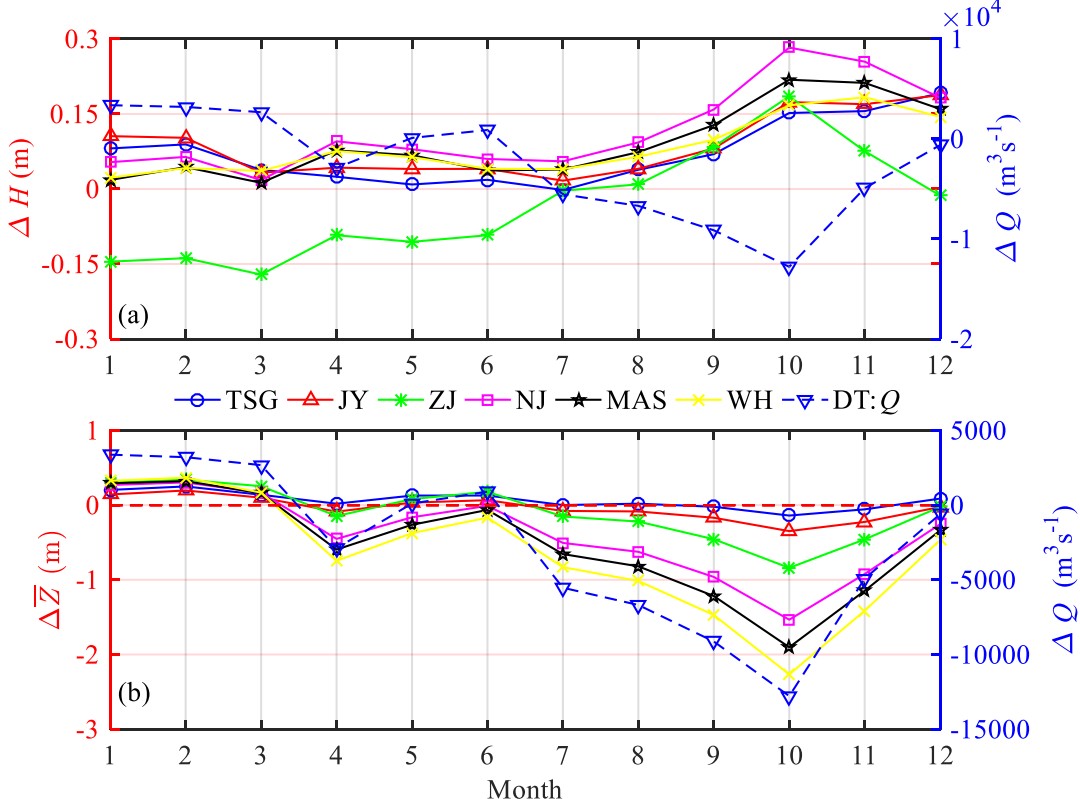

Figure 2. Changes in monthly averaged (a) tidal range $\Delta H$ and (b) residual water level

$\Delta \bar{Z}$ together with the freshwater discharge $\Delta Q$ along the Yangtze River estuary.

Table 2. Comparison of multi-year monthly averaged river discharge $Q$ (m$^3$·s$^{-1}$) between the pre-TGD and the post-TGD periods

| Month | 1 | 2 | 3 | 4 | 5 | 6 | 7 | 8 | 9 | 10 | 11 | 12 |
|---|---|---|---|---|---|---|---|---|---|---|---|---|
| Pre-TGD | 9520 | 10527 | 16298 | 25050 | 30867 | 38283 | 49900 | 47276 | 45317 | 38467 | 23633 | 14810 |
| Post-TGD | 12896 | 13733 | 18974 | 22165 | 30971 | 39180 | 44367 | 40590 | 36187 | 25682 | 18714 | 14203 |
| Change | 3376 | 3206 | 2675 | -2885 | 105 | 896 | -5533 | -6687 | -9130 | -12784 | -4919 | -607 |

Since the TGD operation affects tide-river dynamics primarily through the alteration of the freshwater discharge, it is worth exploring the patterns of trends in the relationship between the freshwater discharge and gradients of the main tidal parameters with respect to distance (i.e. the tidal damping rate and the residual water level slope). Here, we estimated the tidal damping rate $\delta_H$ and the residual water level slope $S$ for a reach of $\Delta x$ by using the following expressions:

$$\delta_H = \frac{1}{(H_1 + H_2)/2} \frac{H_2 - H_1}{\Delta x}, \tag{6}$$

$$S = \frac{\overline{Z_2} - \overline{Z_1}}{\Delta x}, \tag{7}$$

where $H_1$ and $\overline{Z_1}$ are the tidal amplitude and residual water level on the seaward side, respectively, whereas $H_2$ and $\overline{Z_2}$ are the corresponding values $\Delta x$ upstream, respectively. Figure 3 presents the computed tidal damping rates for different reaches along the Yangtze estuary based on the observed tidal ranges at the six gauging stations. It is remarkable that the tidal damping rates at the ZJ-NJ and MAS-WH reaches have significantly increased during the post-TGD period, which suggests an enhancement of tidal dynamics under the current freshwater discharge conditions. On the contrary, a noticeable decrease in $\delta_H$ was observed at the JY-ZJ reach, which corresponds to a

decrease in tidal range at the ZJ station for the low river discharge conditions (from
January to May, see Figure 2a). At TGS-JY and NJ-MAS, no significant change in $\delta_H$
is observed. In Figure 4, a consistent decrease in the residual water level slope $S$ is
observed along the Yangtze estuary, except for the JY-ZJ reach. This means that the
residual friction effect becomes weaker in the post-TGD period since the residual water
level slope is primarily balanced by the residual friction term (Cai et al., 2014a, b, 2016).

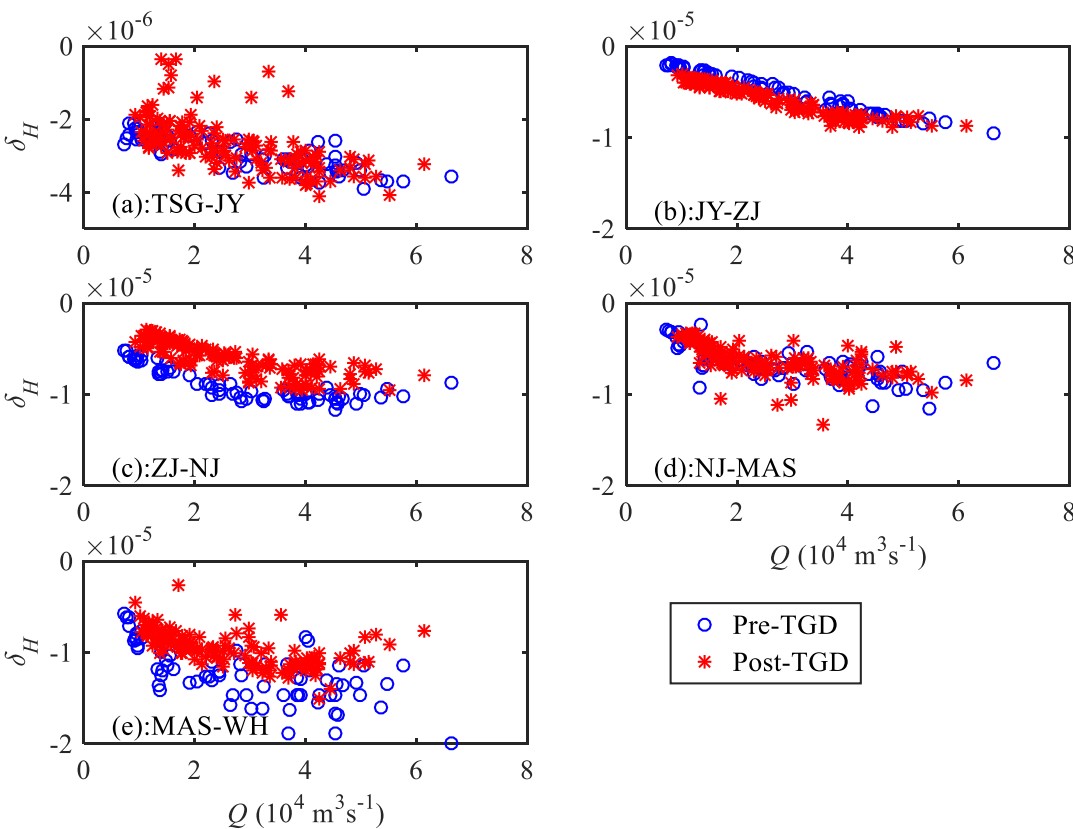


Figure 3. Changes in tidal damping rate $\delta_H$ before and after the TGD closure for
different reaches along the Yangtze estuary: (a) TGS-JY, (b) JY-ZJ, (c) ZJ-NJ, (d) NJ-
MAS, (e) MAS-WH.

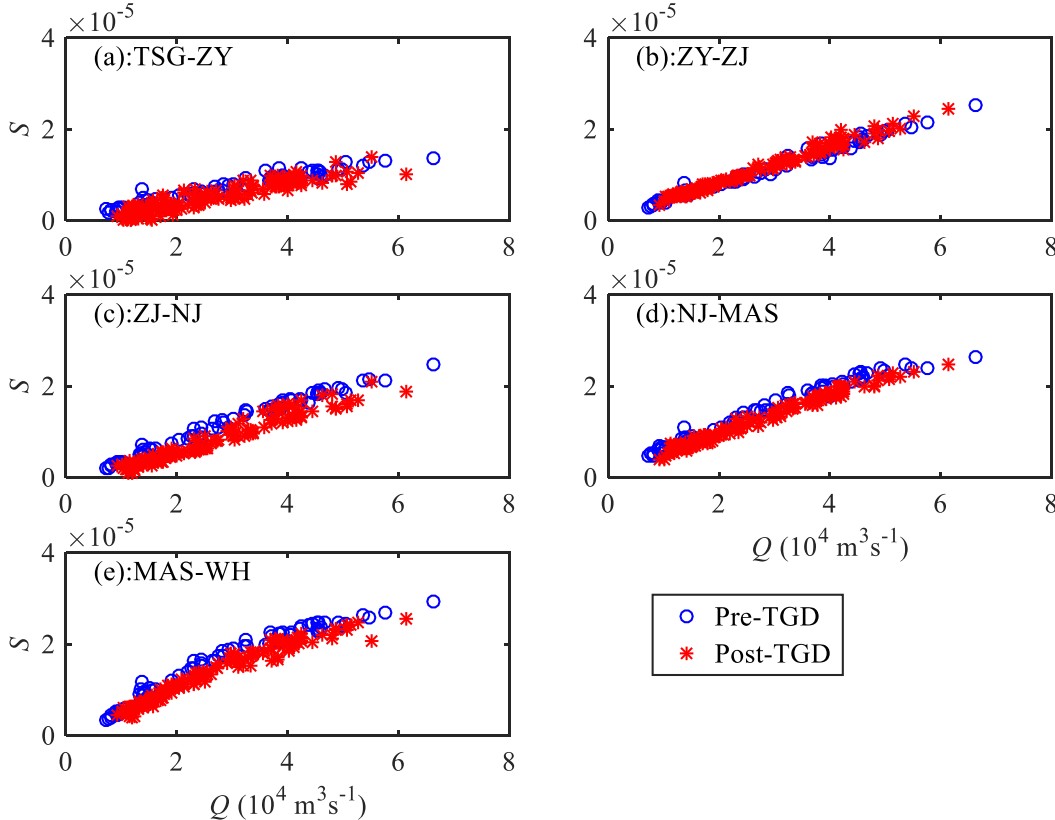

Figure 4. Changes in residual water level slope $S$ before and after the TGD closure for different reaches along the Yangtze estuary: (a) TGS-JY, (b) JY-ZJ, (c) ZJ-NJ, (d) NJ-MAS, (e) MAS-WH.

## 4.2 Performance of the analytical model reproducing the tide-river dynamics

The analytical model presented in Section 3.2 was subsequently applied to the Yangtze River estuary, with the seaward boundary using the tidal amplitude imposed at the TSG station and the landward boundary using the river discharge imposed at the DT station. The computation length of the estuary is 470 km, covering the entire estuary from TSG to DT. The adopted geometric characteristics (including the tidally averaged cross-sectional area, width, and depth) are the same for both pre- and post-TGD periods, which were extracted from a digital elevation model (DEM) using Yangtze River

estuary navigation charts surveyed in 2007. The geometric characteristics, calibrated
by fitting the observed values using Equations (4) and (5), are presented in Table 3,
where a relatively large cross-sectional area convergence length ($a$ = 151 km) is evident,
with a relatively small width ($b$ = 44 km), indicating a fast transition from a funnel-
shaped reach to a prismatic reach in terms of width. It is worth noting that the Yangtze
River estuary is characterised by a typical semidiurnal character; thus, a typical $M_2$ tidal
period (i.e. 12.42 h) was adopted in the analytical model. For the sake of simplification,
we assume that the storage width ratio $r_S$ = 1. Hence, the only calibrated parameter is
the Manning-Strickler friction coefficient $K$. Here, we used two values for $K$: $K$ = 80
$m^{1/3} \cdot s^{-1}$ in the tide-dominated region ($x$ = 0–32 km), and a smaller value of $K$ = 55
$m^{1/3} \cdot s^{-1}$ in the river-dominated region ($x$ = 52–450 km). In addition, to avoid sharp jump
in the analytically computed parameters due to the adoption of different friction
coefficients, we adopted a friction coefficient of $K$=80-55 $m^{1/3}s^{-1}$ (indicating a linear
reduction of the friction coefficient) over the transitional reach ($x$=32-52 km). The
analytically computed results were compared with the observed tidal amplitudes and
the residual water levels at five gauging stations along the Yangtze estuary (Figure 5).
It can be seen that the overall correspondence between analytical results and
observations is good, with high coefficients of determination ($R^2$ > 0.95), which
suggests the usefulness of the present analytical model for reproducing the tide-river
dynamics, given the gross features of flow characteristics and estuarine geometry.


Table 3. Characteristics of geometric parameters in the Yangtze River estuary

| Characteristics | River | Mouth | Convergence length $a/b$ (km) |
|---|---|---|---|
| Cross-sectional area $\bar{A}$ (m$^2$) | 12,135 | 51,776 | 151 |
| Width $\bar{B}$ (m) | 2005 | 6735 | 44 |

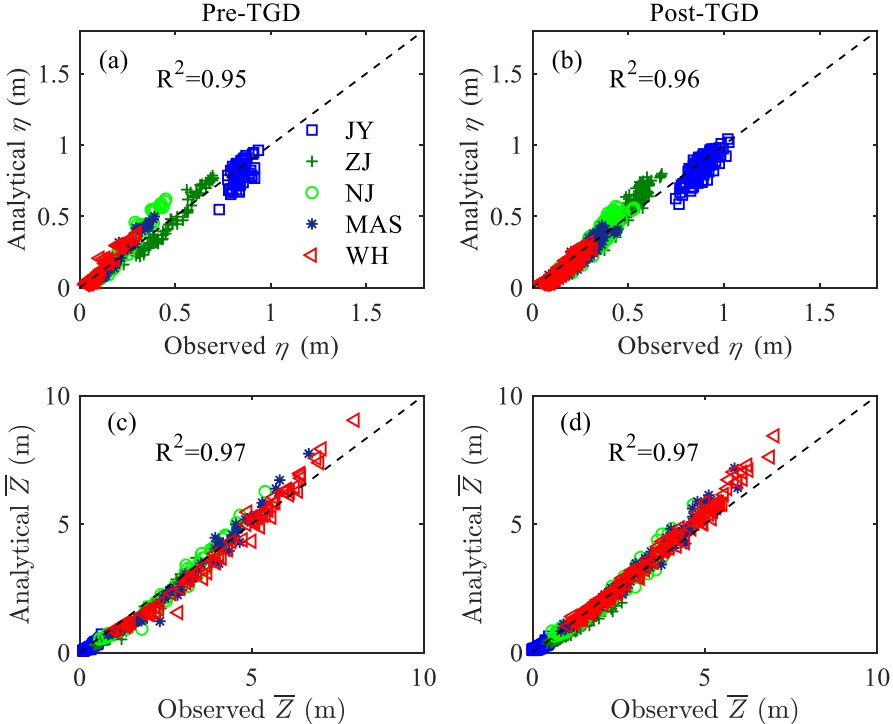

Figure 5. Comparison of monthly averaged values for (a, b) analytically computed tidal amplitude $\eta$ and (c, d) residual water level $\bar{Z}$ against the observations in the Yangtze River estuary for the pre-TGD period (1979–1984) and post-TGD period (2003–2014).

**4.3 Impacts of TGD operation on spatial-temporal patterns of tide-river dynamics**

With the significant seasonal discharge variations resulting from the TGD regulation, an understanding of the seasonal impacts on tide-river dynamics along the estuary has

become increasingly important. In Figures 6 and 7, we see how the TGD operation
impacts the longitudinal variation of the main tidal dynamics in terms of the four
dependent parameters $\delta$, $\lambda$, $\mu$, and $\varepsilon$ for different seasons. The most considerable changes
in the major tide-river dynamics occurred in both autumn and winter seasons, which
correspond to the substantial reduction in freshwater discharge in the wet-to-dry
transition period (i.e. autumn) and slightly increased freshwater discharge in the dry
season (i.e. winter) due to the TGD operation since 2003 (see Table 2). On the other
hand, the impacts of the TGD operation on the tide-river dynamics during the spring
and summer are relatively minor due to the negligible change in the freshwater
discharge. However, we do notice that the TGD had exerted slight influence on tide-
river dynamics in the downstream reaches ($x < 250$ km) during the summer, with the
maximum freshwater discharge occurring within a year. In addition, it appears that there
exists a critical position corresponding to the maximum tidal damping (or minimum
value of $\delta$) upstream in which the tidal damping becomes weak. This phenomenon
occurs particularly in the spring, summer, and autumn. The underlying mechanism is
elaborated in the discussion section.

Figures 6a, c, e, g show the comparison of the analytically computed tidal damping
number $\delta$ before and after the closure of the TGD, in which we clearly observe that the
longitudinal tidal damping effect was considerably weakened in autumn, while it was
slightly enhanced in winter after the TGD closure. This was expected since freshwater
discharges tend to dampen the tidal wave primarily through the enhancement of the
friction term (Horrevoets et al., 2004; Cai et al., 2014a, b, 2016). Figures 6b, d, g, i
show a similar picture for the wave celerity number $\lambda$, which is positively correlated to
the tidal damping number $\delta$, according to the celerity equation (11) in Appendix B.
Figure 7 shows the longitudinal computation of the velocity number $\mu$ and the phase
lag $\varepsilon$ for both periods. The impacts of the TGD operation on the velocity scale and phase
lag are similar to the tidal damping, i.e. the larger the freshwater discharge, the smaller
the velocity number and the phase lag. In Figures 6 and 7, there exist switches of the
analytically computed parameters at both ends of the transitional reach ($x$=32-52 km)
owing to the change in friction coefficient adopted in the analytical model.

Overall, in the seaward reach of the estuary, the effect of freshwater discharge alteration
by the TGD operation on the major tide-river dynamics (i.e. $\delta$, $\lambda$, $\mu$, and $\varepsilon$) was less
significant because of the small ratio of freshwater discharge to tidal discharge. On the
other hand, in the upstream reach of the estuary, the changes in the four dependent
parameters are also small due to the substantial tidal attenuation as a result of the long-
distance propagation from the estuary mouth. Therefore, the pattern of seasonal
variation due to the TGD operation is relatively small at both ends of the estuary,
whereas the largest variation usually occurs in the middle reach of the estuary. This
finding was supported by the results of harmonic analysis using the numerical results
(Zhang et al., 2018). Similar phenomena have also been identified in other large fluvial
meso-tide estuaries, such as the Mekong River estuary and Amazon River estuary,
where dam operation altered the seasonal patterns of tide-river dynamics (Kosuth et al.,
2009; Hecht et al., 2018).

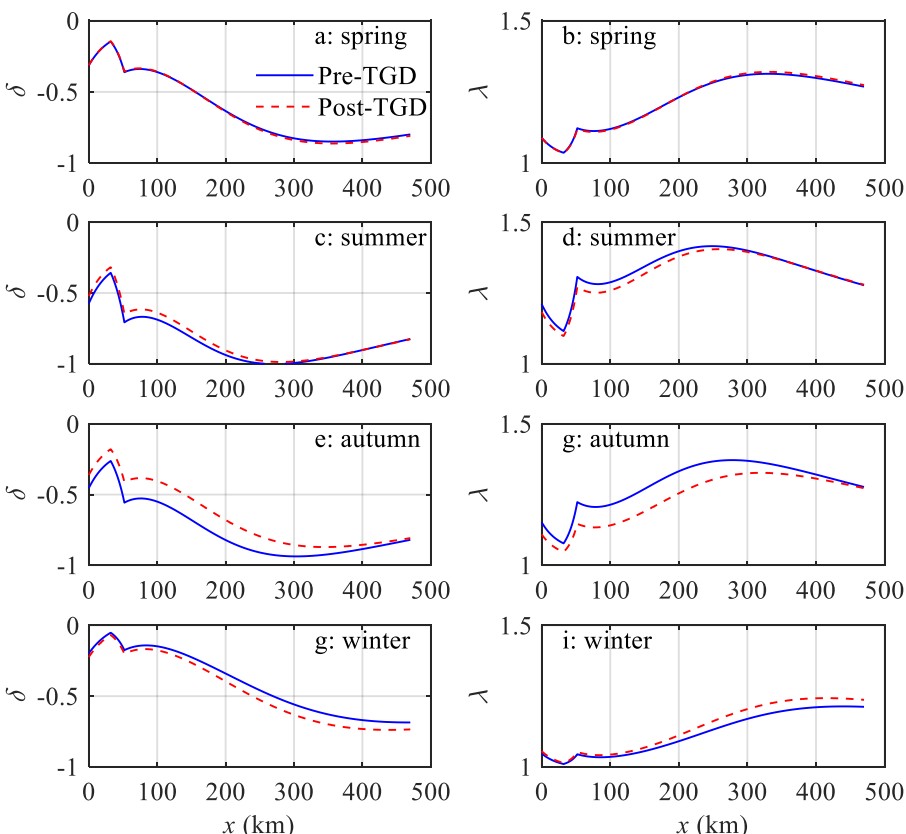

Figure 6. Longitudinal variability of simulated tidal damping number $\delta$ (a, c, e, g) and
celerity number $\lambda$ (b, d, g, i) along the Yangtze estuary in different seasons (spring: a,
b; summer: c, d; autumn: e, g; winter: g, i) for both the pre-TGD and the post-TGD
periods.

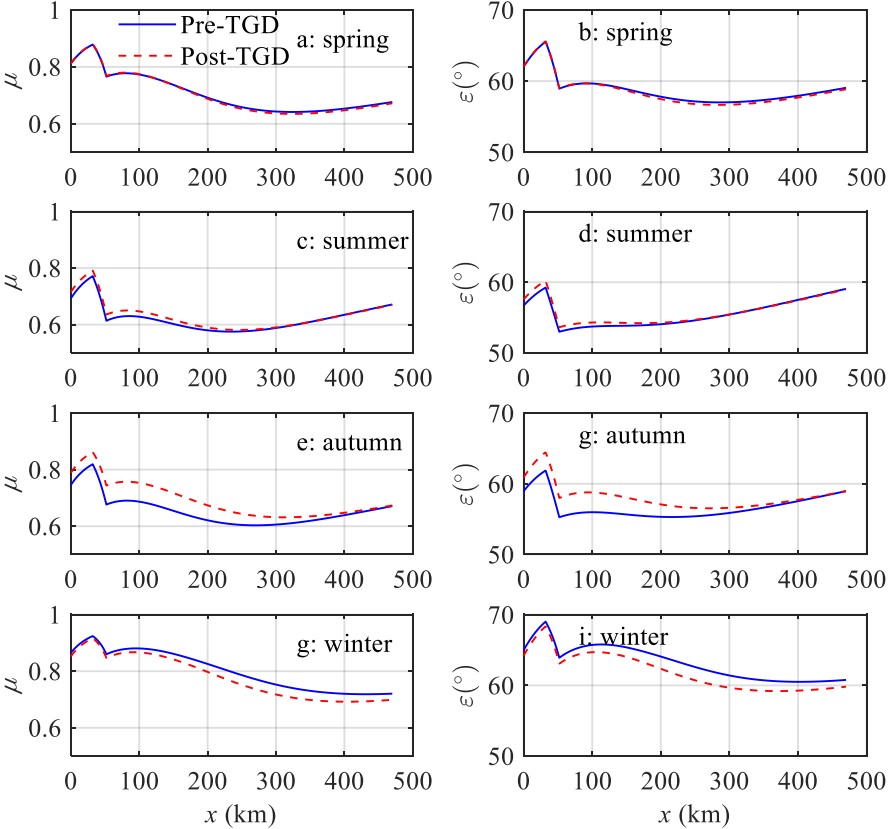


Figure 7. Longitudinal variability of simulated velocity number $\mu$ (a, c, e, g) and phase

lag $\varepsilon$ (b, d, g, i) along the Yangtze estuary in different seasons (spring: a, b; summer: c,

d; autumn: e, g; winter: g, i) for both the pre-TGD and the post-TGD periods.


## 5.  Discussion

### 5.1 The impact of channel geometry alteration on tide-river dynamics

Dam operations, which dramatically modified downstream flow and sediment regimes,

are becoming an increasingly important factor controlling the morphological evolution.

Previous studies show that, as a result of the trapping of sediments by the TGD,

considerable erosion occurred in the first several hundred km downstream of the TGD,

considerably coarsening the bedload (Yang et al., 2014). In particular, the river bed

immediately downstream was eroded at a rate of 65 Mt/yr in 2001–2002 (Yang et al.,
2014). It was shown by Lyu et al. (2018) that due to a dramatic reduction in the sediment
discharge following the construction of the TGD, a significant change in size, geometry,
and spatial distribution of pool-riffles occurred downstream; however, this adjustment
was limited to the reaches close to the TGD. It should be noted that the bathymetry
adopted in the analytical model is restricted to the estuarine area in 2007, which is only
4 years after the TGD closure in 2003, and it is before the full operation of the TGD
began in 2009. In addition, the TGD is around 1600 km away from the estuary mouth,
and its influence on the estuarine morphology normally has a lag effect of at least 4–5
years, as discussed by Wang et al. (2008). Hence, the adopted geometry has been only
partly altered after the TGD closure. The morphological change of Yangtze Estuary can
be even more profound in recent years due to the continuous and accumulated impact
from the TGD. Further adjustment of morphological change due to the sedimentation
in the TGD could exert a considerable impact on the tide-river dynamics in the estuarine
region (e.g., Du et al., 2018; Shaikh et al., 2018). Further study on the impact of
morphological adjustment on the tide-river dynamics is required in the future.

**5.2 The impact of freshwater discharge alteration on tide-river dynamics**
The water conservancy of the TGD has multiple purposes, in which the seasonal
discharge regulation and their impact on the ecosystem are well documented (e.g. Mei
et al., 2015a, b; Chen et al., 2016; Guo et al., 2018). However, the actual influence of
discharge regulation on the river-tide dynamics in the estuarine area is not fully
understood. With the analytical reproduction of tide-river dynamics for pre- and post-
TGD periods, it is possible to quantify the extent of the changes in the major tidal
dynamics, including the estuary shape number $\gamma$ and friction number $\chi$ (Figure 8), and
the residual water level slope $S$ and water depth $h$ (Figure 9) along the Yangtze River
estuary. In general, during the transition from the wet season (summer–autumn) to the
dry season (winter–spring), the water level and corresponding fluvial discharge
downstream from the TGD is first raised by the impounding water and then reduced by
the release of water, which would substantially change the tide-river dynamics in the
downstream estuarine area, with the maximum variation occurring in autumn and the
minimum variation occurring in spring.

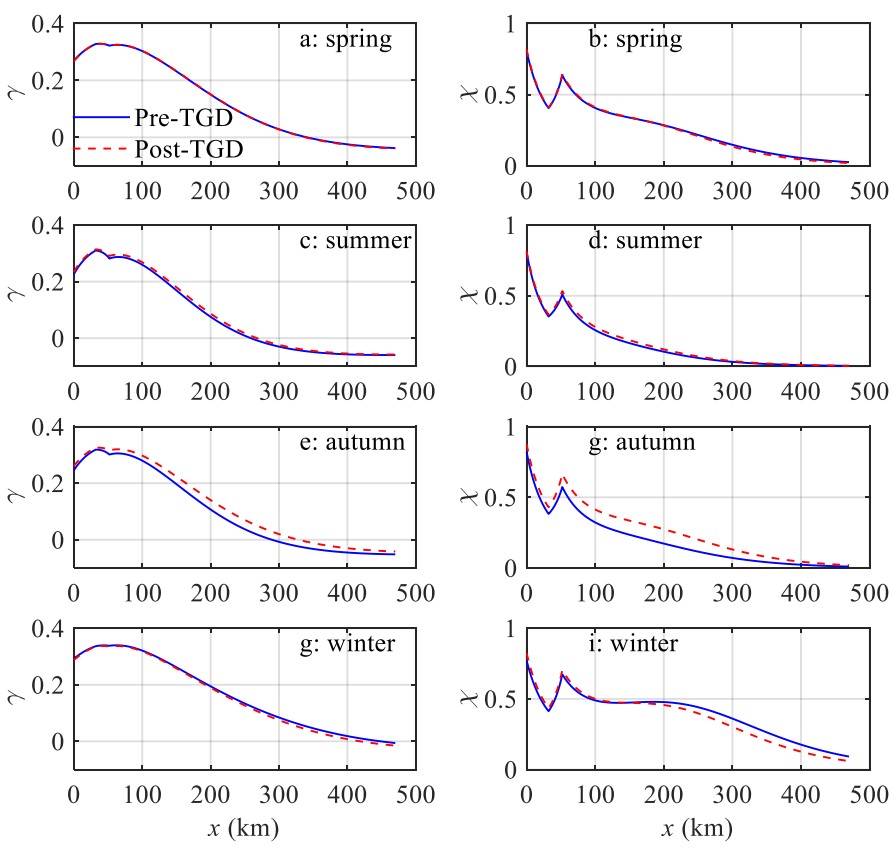


Figure 8. Longitudinal variability of simulated estuary shape number $\gamma$ (a, c, e, g) and
friction number $\chi$ (b, d, g, i) along the Yangtze estuary in different seasons (spring: a, b;
summer: c, d; autumn: e, g; winter: g, i) for both the pre-TGD and the post-TGD periods.

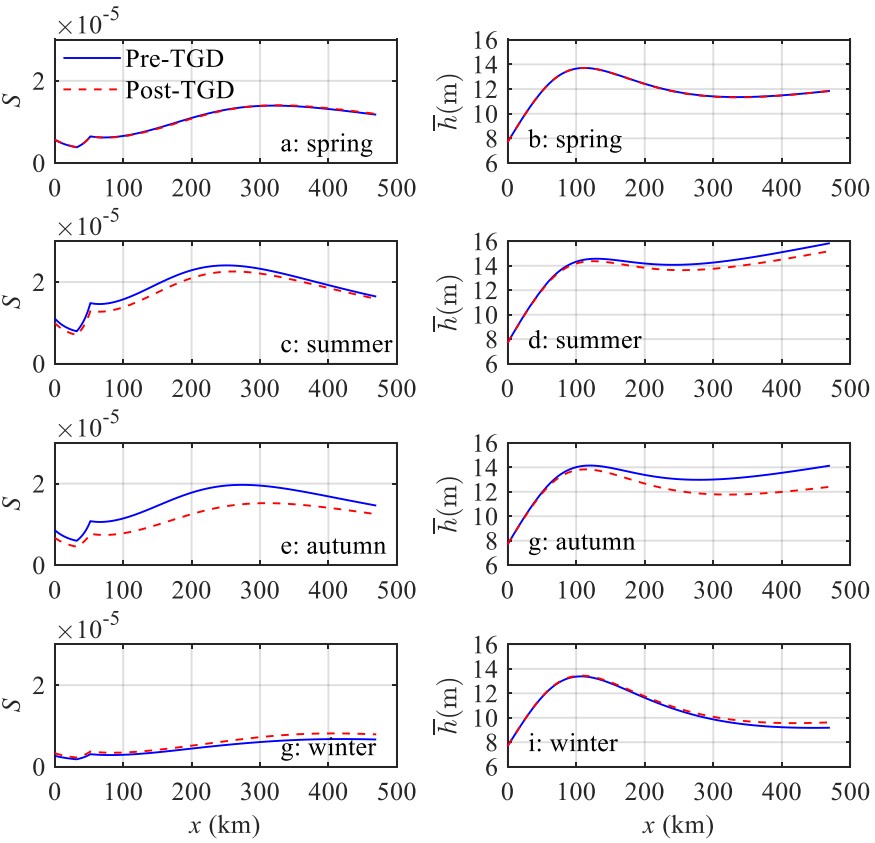


Figure 9. Longitudinal variability of simulated residual water level slope $S$ (a, c, e, g)
and water depth $h$ (b, d, g, i) along the Yangtze estuary in different seasons (spring: a,
b; summer: c, d; autumn: e, g; winter: g, i) for both the pre-TGD and the post-TGD
periods.

Figures 8 and 9 show that during the wet season (summer–autumn), the estuary shape
number $\gamma$ and friction number $\chi$ experience a general increase, while a decrease in the
residual water level slope $S$ and water depth $\bar{h}$ can be identified in the post-TGD

period due to the reduction in freshwater discharge. However, the changes in these major dynamics vary significantly along the channel. Near the estuary mouth, where tidal influence overwhelms the influence from freshwater discharge, the difference is relatively small, as the magnitude of the freshwater discharge is small when compared with that of the tidal discharge. Meanwhile at the upstream reach of the estuary, where the riverine influence dominates that of the tide, the difference is also small due to the attenuation of the tidal wave propagation over a long distance. Consequently, the most significant changes in major tide-river dynamics occurred in the middle reach of the Yangtze River estuary due to the discharge regulation of the TGD during the wet season. By contrast, during the dry season (winter–spring), especially in winter, the opposite trend was observed, indicating a slight increase in $\gamma$ and $\chi$, and a slight decrease in $S$ and $\bar{h}$ due to the additional release of discharge from the TGD. In addition, we also observed that the changes in tide-river dynamics caused by the TGD operation were much stronger upstream than in the lower stream.

**5.3 Implications for water resource management**

The construction of the TGD is the largest hydro-development project ever performed in the world, having multiple influences on downstream water resource management, including navigation, flood control, tidal limit variation, and salt intrusion.

**5.3.1   Implications for navigation**

The navigation condition is mainly controlled by both high water and low water levels.

Figure 10 shows the estimation of the cumulative distribution function (cdf) for both
the high-water level (Figure 10a) and the low-water level (Figure 10b) at the six
gauging stations along the Yangtze River estuary for both the pre- and post-TGD
periods. The results indicate that navigation conditions during the non-flood season are
generally improved, because both percentages of high-water and low-water levels are
increased due to the additional freshwater discharge released from the TGD. On the
other hand, during the flood season, the reduction in the freshwater discharge by TGD
impounding tends to exert a negative impact on navigation. However, the reduced
freshwater discharges in the late summer and autumn are not of sufficient magnitude to
cause any navigation problems. This is due to the fact that the mean water levels during
the flood season are relatively high; hence, the regulating flow quantity and regulating
capacity are relatively small (e.g. Chen et al., 2016). In general, due to the staggered
regulation in freshwater discharge, seasonally, the actual navigation condition is
improved due to the significant increase in the percentage of low water levels.

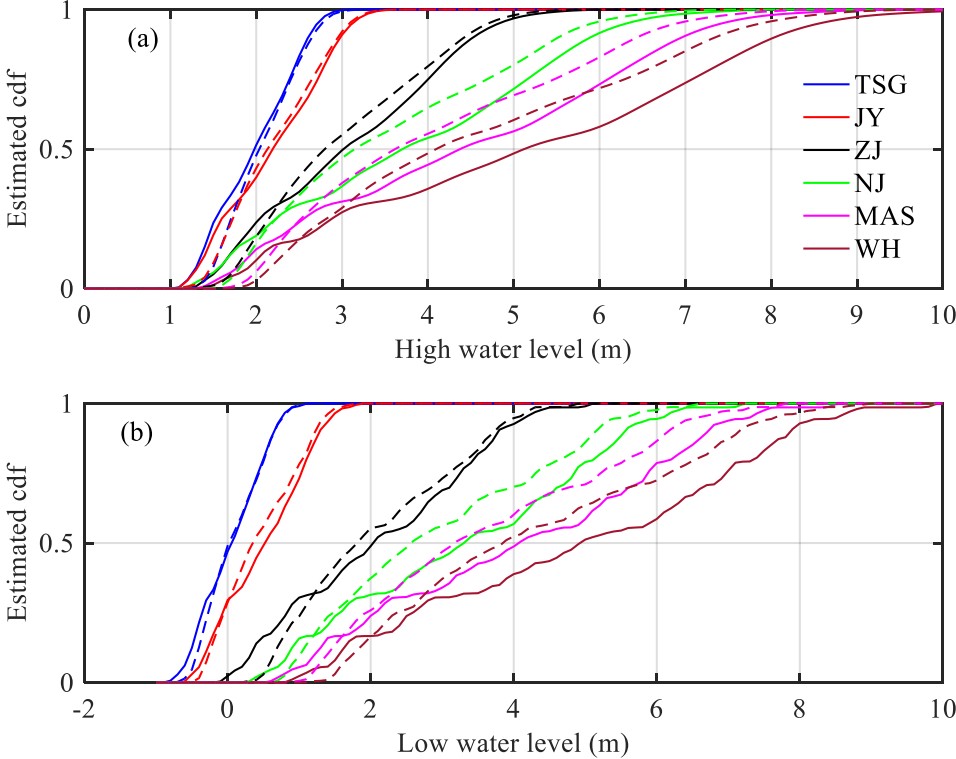

Figure 10. Cumulative distribution function (cdf) estimated by using the kernel smoothing function (a) for high water level and (b) low water level at six gauging stations along the Yangtze estuary. The solid lines represent the pre-TGD period, while the dashed lines represent the post-TGD period.

### 5.3.2 Implications for flood control

Flood control is one of the most important functions of building dams and reservoirs in large rivers. Before the construction of the TGD, the Yangtze River basin suffered from frequent and disastrous flood threats. For instance, the floods of 1998 in the Yangtze River were reported to have killed 3656 people, destroyed 5.7 million homes, and damaged seven million more. Many studies have examined the flood control capacity of the TGD over the past two decades (Zhao et al., 2013; Chen et al., 2014). In particular, the capability of the TGD flood control is influenced by multiple factors (e.g. Huang et

al., 2018), particularly in the estuarine area, which is strongly influenced by tides from the ocean. During the flood season, the reduced freshwater discharge by TGD impounding benefits the flood control by reducing the peak flood discharge. However, as the tidal influence is enhanced, both the percentages of high water and low water levels for the post-TGD period are considerably increased, as shown in Figure 10, indicating a decreased flood control capability. For instance, at the WH gauging station located in the upstream part of the Yangtze River estuary, the 8-m high-water level increased by approximately 10% after the TGD closure during the wet season. The corresponding flood prevention standard, therefore, is reduced due to the increased high-water level (see also Nakayama and Shankman, 2013).

### 5.3.3  Implications for tidal limit

It is important to detect the position of the tidal limit (corresponding with the position where the tidal amplitude to depth ratio is less than a certain threshold, e.g. $\frac{\eta}{h} < 0.02$), which is the farthest point upstream where a river is affected by tidal fluctuations, since it is essential for surveying, navigation, and fisheries management, in general (e.g. Shi et al., 2018). Subsequently, we are able to define the tide-influenced length as the distance upstream from the estuary mouth to the tidal limit. Generally, the tidal limit fluctuates with the changes in the seasonal freshwater discharges. Field measurements have demonstrated that tidal limit can reach as far as the NJ station and further upstream during the dry season, while during the wet season, it is pushed down to the ZJ station and may be pushed further downward to the JY station under spate conditions. Figure 11 shows the analytically computed tidal limit position for both the pre- and post-TGD

periods. It can be observed that the tidal limit moved downstream by about 45 km and 39 km in January and February under the impact of the additional release of discharge from TGD during the dry season. During the transition from dry to wet seasons (January–May), the total freshwater discharge from TGD increases, and we identify further downstream movement of the tidal limit, although to a smaller extent. The reverse of the post-TGD tidal limit in April is due to the decrease in the freshwater discharge compared with the pre-TGD tidal limit (see Table 2). The TGD storage period begins in June, and the tidal limit moved upstream by a large amount compared with the pre-TGD period. The largest change occurred during October when the tidal limit moved from 175 km pre-TGD to 250 km post-TGD due to the substantial increase in freshwater discharge (see Table 2).

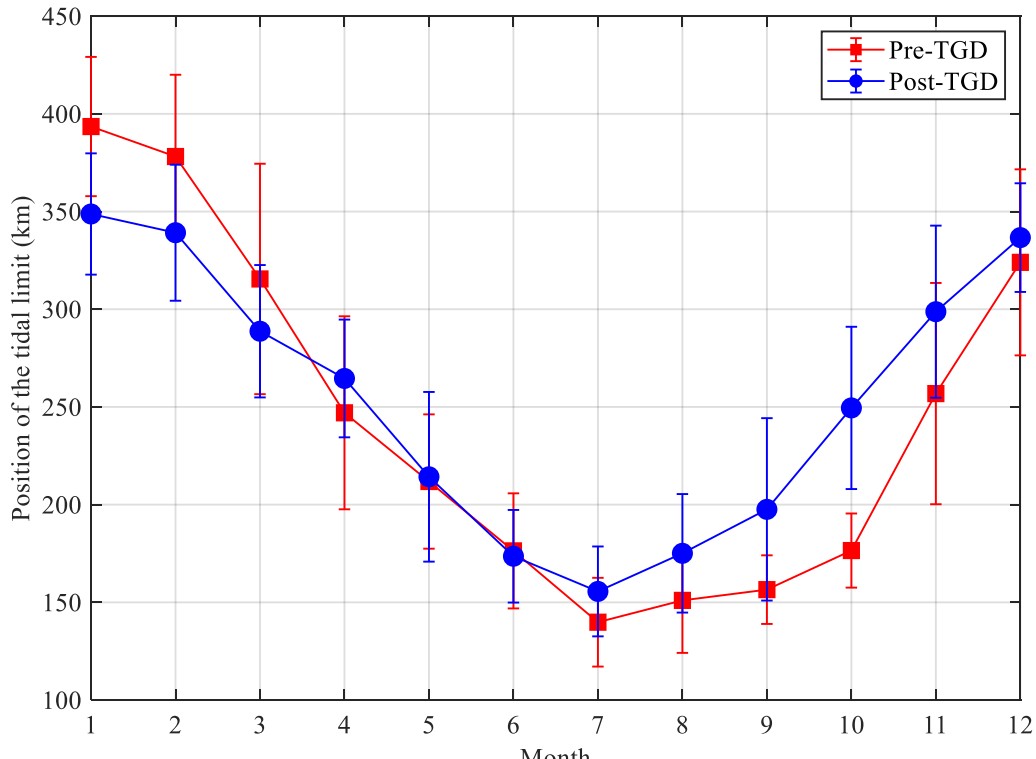

Figure 11. Temporal variation of the position of the tidal limit relative to the TSG station
for both the pre-TGD and the post-TGD periods. The vertical error bar at each data
point indicates the standard deviation of the analytically computed time series.

### 5.3.4 Implications for salt intrusion

The operation of the TGD changed the location of tidal limit, which, in turn, directly
influences the intensity of saltwater intrusion, especially during the dry season, when
the freshwater discharge is low and saltwater intrusion is important (e.g. Cai et al.,
2015). The analysis of tide-river dynamics shows that the tidal dynamics are
considerably enhanced during the autumn due to the substantial decline in freshwater
discharged into the estuary, which may lead to enhanced saltwater intrusion. However,
with supplemented discharge after the TGD during the winter, saltwater intrusion tends
to be significantly suppressed, and the isohalines are pushed seaward by additional river
discharges (e.g. An et al., 2009; Qiu and Zhu, 2013). In contrast, during the wet season,
the TGD operation slightly extended the timing of saltwater intrusion and increased its
intensity by impounding freshwater. Since the total river discharge rate during the wet
season is the largest during the year, the influence of saltwater on freshwater reservoirs
along the coastal area is limited. Therefore, the operation of TGD is overall favourable
for reducing the burden of freshwater supplement in the tidally influenced estuarine
areas. However, to quantify the potential impacts of TGD's operation on salt intrusion
and related aquatic ecosystem health in general, it is required to couple the
hydrodynamic model to the ecological or salt intrusion model (e.g., Qiu and Zhu, 2013;
Cai et al., 2015).

## 6.   Conclusions

An analytical approach was used to examine the potential impacts of TGD operation
on the spatial-temporal patterns of tide-river dynamics along the Yangtze River estuary.
It was shown that the freshwater regulation caused by the TGD, on a seasonal scale,
exerts significant impacts on the tide-river dynamics, with the maximum influence
occurring in autumn and winter. This generally corresponds to a dramatic decrease in
freshwater discharge during the wet-to-dry transition period and a slight increase in
discharge during the dry season. The analytical results indicate that the discharge
regulation by the TGD drives the alterations in the tide-river dynamics instead of the
geometric change. In particular, the change in the freshwater discharge changes the
estuary shape number (representing the geometric effect), the residual water level slope
(representing the effective frictional effect) and, hence, the tide-river dynamics. This
study, using the Yangtze River estuary as an example, provides an effective yet simple
method to quantify the seasonal regulation in freshwater discharge by large reservoirs
or dams on hydrodynamics in estuaries. The results obtained from this study will,
hopefully, shed new light on aspects of water resource management, such as navigation,
flood control, and salt intrusion.

**Data availability.** Data and results are available from the authors upon request.
**Author contributions.** All authors contributed to the design and development of the

work. The experiments were originally carried out by Huayang Cai. Xianyi Zhang and

Leicheng Guo carried out the data analysis. Min Zhang built the model and wrote the

paper. Feng Liu and Qingshu Yang reviewed the paper.

**Competing interests.** The authors declare that they have no conflict of interest.

**Acknowledgments.** We acknowledge the financial support from the National Key R&D of China (Grant No. 2016YFC0402600), from the Open Research Fund of State Key Laboratory of Estuarine and Coastal Research (Grant No. SKLEC-KF201809), from the National Natural Science Foundation of China (Grant No. 51709287 and 41701001), and from the Guangdong Provincial Natural Science Foundation of China (Grant No. 2017A030310321).

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

**Appendix A. Simplified momentum balance for the residual water level slope**

Assuming a periodic variation of flow velocity, the integration of Equation (1) over a
tidal cycle leads to an expression for the residual water level slope (e.g. Cai et al., 2014a,

776    2016):

$$\frac{\partial \overline{Z}}{\partial x} = -\frac{1}{K^2}\overline{\left(\frac{U|U|}{h^{4/3}}\right)} - \frac{1}{2g}\frac{\partial \overline{U^2}}{\partial x} - \frac{1}{2\rho_0}\overline{h\frac{\partial \rho}{\partial x}} \qquad (8)$$

where the overbars and the subscript 0 indicate the tidal average and value at the
seaward boundary, respectively. The residual water level slope is induced by three
contributions: residual frictional, advective acceleration, and density effects, which
correspond to the three terms on the right-hand side of Equation (8). Note that the
contribution from advective acceleration to the residual water level slope:

$$\frac{\partial \overline{Z}_{adv}}{\partial x} = -\frac{1}{2g}\frac{\partial \overline{U^2}}{\partial x}, \qquad (9)$$

can be easily integrated to:

$$\overline{Z}_{adv} = -\frac{1}{2g}\left(\overline{U^2} - \overline{U_0^2}\right) = -\frac{1}{2}\overline{Fr_0}\left(\frac{\overline{U^2}}{\overline{U_0^2}} - 1\right)\overline{h_0} \qquad (10)$$

where the Froude number is introduced, $\overline{Fr^2} = \overline{U^2}/(g\overline{h})$, which is computed with the
averaged variables. In this case, the correction is local (not cumulative) and
proportional to the flow depth through a coefficient that is negligible as long as the
velocity does not change significantly, and $Fr$ is small, as is common in most tidal flows.
It was shown by Savenije (2005, 2012) that the density term in equation (1) always
exercises a pressure in the landward direction, which is counteracted by a residual water
level slope, amounting to 1.25% of the estuary depth over the salt intrusion length. The
value for the residual water level slope, induced by the density effect, is usually small
compared with the gradient of the free surface elevation; thus, in this paper, we neglect
the influence of the density difference on the dynamics of the residual water level.

**Appendix B. Governing equations for tide-river dynamics in estuaries**
The analytical solutions for the dependent parameters $\mu$, $\delta$, $\lambda$, and $\varepsilon$ are obtained by
solving the following four dimensionless equations (see details in Cai et al., 2014a):
the tidal damping/amplification equation, describing the tidal amplification or damping
as a result of the balance between channel convergence $\left(g q\right)$ and bottom friction
$\left(c m / \mathrm{G}\right)$:

$$\delta = \frac{\mu^2\left(\gamma\theta - \chi\mu\lambda\Gamma\right)}{1 + \mu^2\beta}, \tag{11}$$


the scaling equation, describing how the ratio of velocity amplitude to tidal amplitude
depends on phase lag and wave celerity:

$$\mu = \frac{\sin\left(\varepsilon\right)}{\lambda} = \frac{\cos\left(\varepsilon\right)}{\gamma - \delta}, \tag{12}$$


the celerity equation, describing how the wave celerity depends on the balance between
convergence and tidal damping/amplification:
$$\lambda^2 = 1 - \delta(\gamma - \delta),\tag{13}$$

and the phase lag equation, describing how the phase lag between HW and HWS
depends on wave celerity, convergence, and damping:
$$\tan(\varepsilon) = \frac{\lambda}{\gamma - \delta},\tag{14}$$

where $q$, $b$, and $G$ account for the effect of river discharge and where:
$$\beta = \theta - r_s\zeta\varphi/(\mu\lambda), \quad \theta = 1 - \left(\sqrt{1+\zeta} - 1\right)\varphi/(\mu\lambda), \quad \Gamma = \frac{1}{\pi}\left[p_1 - 2p_2\varphi + p_3\varphi^2\left(3 + \mu^2\lambda^2/\varphi^2\right)\right].$$

815                                      (15)

Note that $\Gamma$ is a friction factor obtained by using Chebyshev polynomials (Dronkers,
1964) to represent the non-linear friction term in the momentum equation:
$$F = \frac{U|U|}{K^2\overline{h}^{4/3}} \approx \frac{1}{K^2\overline{h}^{4/3}} \frac{1}{\pi}\left(p_0\upsilon^2 + p_1\upsilon U + p_2 U^2 + p_3 U^3/\upsilon\right)\tag{16}$$

in which $U$ is the cross-sectional averaged velocity consisting of a steady component
$U_r$, generated by the fresh water discharge, and a time-dependent component $U_t$,
introduced by the tide:
$$U = U_t - U_r = \upsilon\sin(\omega t) - Q/\overline{A}\tag{17}$$

where $Q$ is the fresh water discharge (treated as a constant during the tidal wave
propagation), and $p_i$ ($i = 0, 1, 2, 3$) are the Chebyschev coefficients (see Dronkers, 1964,
p. 301), which are functions of the dimensionless river discharge $\varphi$ through $\alpha = \arccos(-$
$\varphi$):
$$p_0 = -\frac{7}{120}\sin(2\alpha) + \frac{1}{24}\sin(6\alpha) - \frac{1}{60}\sin(8\alpha),\tag{18}$$

$$p_1 = \frac{7}{6}\sin(\alpha) - \frac{7}{30}\sin(3\alpha) - \frac{7}{30}\sin(5\alpha) + \frac{1}{10}\sin(7\alpha),\tag{19}$$

$$p_2 = \pi - 2\alpha + \frac{1}{3}\sin(2\alpha) + \frac{19}{30}\sin(4\alpha) - \frac{1}{5}\sin(6\alpha),\tag{20}$$

$$p_3 = \frac{4}{3}\sin(\alpha) - \frac{2}{3}\sin(3\alpha) + \frac{2}{15}\sin(5\alpha).\tag{21}$$

The coefficients $p_1$, $p_2$, and $p_3$ determine the magnitudes of the linear, quadratic, and
cubic frictional interaction, respectively.