# Peer review of "Impacts of Three Gorges Dam's operation on spatial-temporal"

_Ocean Science, 2018_

## Referee Comment (RC1) · Du (Referee) · 3 Feb 2019

The article used an analytical tidal model to understand the change of freshwater discharge on the tidal dynamics in the Yangtze River, with a specific focus on the impact of Three Gorge Dam. While freshwater discharge's effect on tidal dynamics is well recognized by observations and numerical models, it is rare to use an analytical tool to understand the underlying mechanism (e.g., bottom friction, tidal damping) for the changes in tidal dynamics. I believe the article is a good example study for the influence of dam construction. There are some issues, however, needed to be well resolved before acceptance for publication.

[Figure]

Major comments

1. The organization of the paper can be improved. For example, in section 4.1, the description of changed tidal amplitude and mean water level is followed by the analytical analysis of the tidal damping before detailing the model performance. I would suggest moving the later part in section 4.1 to section 4.2. Following a strategy as "observational analysis; model performance and validation; analysis of TGD's influence based on the model results".

2. The wording can be greatly improved. Some sentences have been mentioned again and again. For example, similar sentences as in L154-155 "we mainly concentrate on the tide-river dynamics under the impacts of TGD seasonal regulation over the entire reach of the Yangtze River estuary" can be found in multiple places, in the introduction, methods, results. Please revise them and make the text more concise. I would suggest mentioning such a sentence in the introduction and in the conclusion while avoiding repeating them in methods and results. Extensive minor grammar suggestions can be found in the minor comments.

3. While I agreed that the discharge regulation affects the tidal dynamics, I am not convinced that the influence of geometric or morphological change due to TGD is limited. The authors used 2007 bathymetric data, which might not reflect the alteration due to TGD considering the time-lag of 4-5 yrs in morphological response to the TGD. The morphological change can be more profound in recent years and it is well known that the reduced sediment delivery due to the trapping of TGD affects the erosion/deposition status of the Yangtze River delta. It is possible that the morphological change on the tidal might be less profound compared to the river discharge, but such a conclusion is not supported by the presented analysis. I would suggest rewording the related sentence regarding the influence of morphological change.

4. Regarding the TGD's influence on damping rate as shown in Fig. 3, could you explain why there so many jumping values (e.g., in Fig. 3a,c,d,e)? For an analytical

solution with so many simplification assumptions, the response shall be in a much smoother way. Is such a jumping pattern observed in reality? I think it is important to clarify such abnormal features in your figures. Such types of not explained pattern also exist for figure 6-9, where there is a clear jumping pattern. Is it because you are using two manning coefficients for different regions?

5. As the major focus of this paper is to use the model to quantify the impact of freshwater discharge. It is vitally important to show the model can reproduce the change in tidal dynamics (e.g., tidal range) in response to varying freshwater discharge. For example, a plot showing the observed change of tidal range (using the amplitude of M2 would be better) as a function of freshwater input at selected stations, together with another line showing the modeled amplitude as a function of river discharge.

6. For the captions of many figures, it is necessary to detail what each data point represent for. For example, in Figure 5, it not clear to me how each data point is obtained, is it monthly mean value?

Minor comments

L44: "to the extent" here reads awkward. Please consider to revise it.

L54: a recent work by Du et al. (2018) might be a good reference in concern of the geomorphic constraints on tidal dynamics.

Du et al. (2018-GRL). Tidal response to sea‐level rise in different types of estuaries: the importance of length, bathymetry, and geometry

L55: suggest to change "including spring-neap tidal fluctuations as well as seasonal varying discharge" to "in timescale ranging from a fortnight to season"

L57: change "of the river" to "of a river"

L58: delete "being", only those that have already been built can cause changes in downstream freshwater discharge

L64: suggest moving the part "such as xxxx" forward to as "human intervention, such as xxxx, which are xxxx"

L68: suggest changing "a large river" to "the largest river in China in terms of mean discharge" to emphasize the importance of Yangtze River.

L92: suggest changing "that have been mainly been concerned with" to "on", making it more concise.

L114: change "the TGD seasonal regulation effect" to "the effect of TGD seasonal regulation"

L119, L121: suggest using past tense of phase, to be consistent to the phase you used at the beginning, where you used "adopted".

L138: change "Downstream of" to "Downstream"

L143: change "discharge" to "was discharged"

L148: change "a tidal range that extends up to ∼4.6m" to "a tidal range of up to 4.6m"

L157: Delete "Sketch", it is actually a map, not a sketch one. Suggest changing "displaying the location of gauging and hydrological stations" to "with the location of tidal gauging and hydrological stations shown with black solid circles and rec solid rectangles".

L168: change "difference of " to "difference between"

L169: "and a half" to "and dividing by two".

L170: "water levels of xx stations" to "water level at xx stations"

L206: will the solution for rectangle lateral shape channel be different with those with a V-shape? It is better to state here why such an assumption is valid as most part of estuary is not rectangle shape but v-shape

L214-217: These symbols are not used for the four number and it is not appropriate

to use "where" here. I suggest to move it as a note under the table 1, or express the formula for each number explicitly in the text (say, each number is described with its corresponding formula in the text).

L280: "in upstream stations" to "at upstream stations"

L302-309: Isn't it necessary to describe whey some segments has seen little change or even decrease? It is confusing. Why increased damping denotes weaker friction? For classic understanding, it is thought larger friction lead to a higher damping rate.

Figure 5: what does each data point stand for? Monthly value? Or yearly value?

Section 4.3: the second part in section 4.1 is suggested to move into section 4.3.

L373: "identify" seems not a good word here.

L383: in "the larger the freshwater discharge is, the smaller the velocity number and the phase lag are.", suggest changing as "the larger the freshwater discharge, the smaller the velocity number and the phase lag."

Figure 6: why there is sharply jumping in the curve, due to different manning coefficient?

L416: "the river immediately downstream eroded" to "the river bed immediately downstream was eroded"

L426: "therefore" may be a better word than "consequently"

L463-466: this whole sentence reads awkward. Suggesting changing "where the tidal influence dominates that of the freshwater discharge" to "where tidal influence overwhelms the influence from freshwater discharge".

L500: "drawn lines" to "solid lines"

L519: it is not clear how you determine the value "20-yr" and "10-yr" here.

L574: suggest changing "as a significant case study" to "as an example"

---

## Author Comment (AC1) · 21 Feb 2019

**Responses to comments by Reviewer #1**

We thank Dr. Du's careful consideration of our work. In this rebuttal, we have addressed all the comments formulated by the Reviewer by replying (in blue) to his remarks (in black).

**General comments:**

The article used an analytical tidal model to understand the change of freshwater discharge on the tidal dynamics in the Yangtze River, with a specific focus on the impact of Three Gorge Dam. While freshwater discharge's effect on tidal dynamics is well recognized by observations and numerical models, it is rare to use an analytical tool to understand the underlying mechanism (e.g., bottom friction, tidal damping) for the changes in tidal dynamics. I believe the article is a good example study for the influence of dam construction. There are some issues, however, needed to be well resolved before acceptance for publication.

Our reply: Thanks a lot for the positive assessment of our paper.

**Major comments**

1. The organization of the paper can be improved. For example, in section 4.1, the description of changed tidal amplitude and mean water level is followed by the analytical analysis of the tidal damping before detailing the model performance. I would suggest moving the later part in section 4.1 to section 4.2. Following a strategy as "observational analysis; model performance and validation; analysis of TGD's influence based on the model results".

Our reply: We do agree with the strategy you proposed. However, it is worth noting that the later part in section 4.1 actually described the observed tidal damping rate and residual water level slope based on the observed water levels in a monthly scale. Hence the whole section 4.1 only consists of observational analysis without including analytical analysis based on the analytical results. In the revised paper, we shall maintain the same structure of the paper.

2. The wording can be greatly improved. Some sentences have been mentioned again and again. For example, similar sentences as in L154-155 "we mainly concentrate on the tide-river dynamics under the impacts of TGD seasonal regulation over the entire reach of the Yangtze River estuary" can be found in multiple places, in the introduction, methods, results. Please revise them and make the text more concise. I would suggest mentioning such a sentence in the introduction and in the conclusion while avoiding repeating them in methods and results. Extensive minor grammar suggestions can be found in the minor comments.

Our reply: Many thanks for pointing this out. In the revised paper, we shall remove the repeated sentences in the methods and results as suggested by the reviewer.

3. While I agreed that the discharge regulation affects the tidal dynamics, I am not convinced that the influence of geometric or morphological change due to TGD is

limited. The authors used 2007 bathymetric data, which might not reflect the alteration due to TGD considering the time-lag of 4-5 yrs in morphological response to the TGD. The morphological change can be more profound in recent years and it is well known that the reduced sediment delivery due to the trapping of TGD affects the erosion/deposition status of the Yangtze River delta. It is possible that the morphological change on the tidal might be less profound compared to the river discharge, but such a conclusion is not supported by the presented analysis. I would suggest rewording the related sentence regarding the influence of morphological change.

Our reply: We very much appreciate your comment with regard to potential impacts of the morphological adjustment (e.g., loss of floodplain storage by erosion) caused by the construction of the TGD on the tide-river dynamics. Indeed, the relative magnitude of morphological adjustment is likely to progressively increase due to the time-lag of morphological response and the rapid sedimentation in the reservoir (see Mei et al., 2018). Due to the lack of detailed bathymetry data before and after the TGD's operation, in this study we could not further analyze the impacts of morphological adjustment on the tide-river dynamics. In the revised paper, we shall rewrite the sentence concerning the impacts of morphological change: "*The morphological change of Yangtze Estuary can be even more profound in recent years due to the continuous and accumulated impact from the TGD. Further adjustment of morphological change due to the sedimentation in the TGD could exert a considerable impact on the tide-river dynamics in the estuarine region (e.g., Du et al.2018; Shaikh et al., 2018). Further study on the impact of morphological adjustment on the tide-river dynamics is required in the future.*"

4. Regarding the TGD's influence on damping rate as shown in Fig. 3, could you explain why there so many jumping values (e.g., in Fig. 3a,c,d,e)? For an analytical solution with so many simplification assumptions, the response shall be in a much smoother way. Is such a jumping pattern observed in reality? I think it is important to clarify such abnormal features in your figures. Such types of not explained pattern also exist for figure 6-9, where there is a clear jumping pattern. Is it because you are using two manning coefficients for different regions?

Our reply: Figure 3 shows the observed tidal damping rate $\delta_H$ before and after the TGD closure for different reaches along the Yangtze estuary. These values are observations according to Equation (6) in the manuscript, rather than analytical results. The main reason that there exists a jumping pattern lies in the fact that the tidal damping rate $\delta_H$ are very small values (in terms of magnitude of $-10^{-6}$—$10^{-5}$), thus small changes in observed tidal range would dramatically change the damping rate. In the revised paper, we shall explicitly mention about this.

For the jumping behavior in Figures 6-9, this has to do with the adoption of two very different Manning-Strickler friction coefficient in the seaward and landward regions. In the revised paper, to avoid such a sharp jump in the curves and to improve the model performance, we will adopt a friction coefficient of $K$=80-55 m$^{1/3}$s$^{-1}$ (indicating

a linear reduction of the friction coefficient) over the transitional reach ($x$=32-52 km) in the analytical model. Figures R1-R4 below show the updated Figures 6-9 using a linear reduction of the friction coefficient in the transitional reach, where we observe a smooth transition of the analytically computed variables.

[Figure]

Figure R1. Longitudinal variability of simulated tidal damping number $\delta$ (a, c, e, g) and celerity number $\lambda$ (b, d, g, i) along the Yangtze estuary in different seasons (spring: a, b; summer: c, d; autumn: e, g; winter: g, i) for both the pre-TGD and the post-TGD periods.

[Figure]

Figure R2. Longitudinal variability of simulated velocity number $\mu$ (a, c, e, g) and phase lag $\varepsilon$ (b, d, g, i) along the Yangtze estuary in different seasons (spring: a, b; summer: c, d; autumn: e, g; winter: g, i) for both the pre-TGD and the post-TGD periods.

[Figure]

Figure R3. Longitudinal variability of simulated estuary shape number $\gamma$ (a, c, e, g) and friction number $\chi$ (b, d, g, i) along the Yangtze estuary in different seasons (spring: a, b; summer: c, d; autumn: e, g; winter: g, i) for both the pre-TGD and the post-TGD periods.

[Figure]

Figure R4. Longitudinal variability of simulated residual water level slope $S$ (a, c, e, g) and water depth $h$ (b, d, g, i) along the Yangtze estuary in different seasons (spring: a, b; summer: c, d; autumn: e, g; winter: g, i) for both the pre-TGD and the post-TGD periods.

5. As the major focus of this paper is to use the model to quantify the impact of freshwater discharge. It is vitally important to show the model can reproduce the change in tidal dynamics (e.g., tidal range) in response to varying freshwater discharge. For example, a plot showing the observed change of tidal range (using the amplitude of M2 would be better) as a function of freshwater input at selected stations, together with another line showing the modeled amplitude as a function of river discharge.

Our reply: As we mentioned in the introduction part, the main purpose of this paper lies in quantifying the impacts of TGD's seasonal regulation on the tide-river dynamics over the entire reach of the Yangtze River estuary. Concerning the impacts of varying freshwater discharge on the tide-river dynamics, the reviewer can kindly refer to our recent publication in the journal of Hydrology and Earth System Sciences (in discussion): Cai, H., Savenije, H. H. G., Garel, E., Zhang, X., Guo, L., Zhang, M., Liu, F., and Yang, Q., 2018. Seasonal behaviour of tidal damping and residual water level slope in the Yangtze River estuary: identifying the critical position and river discharge for maximum tidal damping, Hydrol. Earth Syst. Sci. Discuss.,

https://doi.org/10.5194/hess-2018-524, in review.

6. For the captions of many figures, it is necessary to detail what each data point represent for. For example, in Figure 5, it not clear to me how each data point is obtained, is it monthly mean value?

Our reply: These are monthly averaged values. In the revised paper, we shall clarify the plotted data in the captions of all the figures. For instance, we shall modify the caption of the Figure 5 as "*Comparison of monthly averaged values for (a, b) analytically computed tidal amplitude η and (c, d) residual water level $\bar{Z}$ against the observations in the Yangtze River estuary for the pre-TGD period (1979–1984) and post-TGD period (2003–2014).*"

**Minor comments**

L44: "to the extent" here reads awkward. Please consider to revise it.
Our reply: We shall replace "to the extent that" with "so that" in the revised paper.

L54: a recent work by Du et al. (2018) might be a good reference in concern of the geomorphic constraints on tidal dynamics.
Our reply: Thank you for pointing this out. In the revised paper, we shall include this recent work.

L55: suggest to change "including spring-neap tidal fluctuations as well as seasonal varying discharge" to "in timescale ranging from a fortnight to season"
Our reply: we agree with your suggestion.

L57: change "of the river" to "of a river"
Our reply: We agree with your comment.

L58: delete "being", only those that have already been built can cause changes in downstream freshwater discharge.
Our reply: We agree with your comment.

L64: suggest moving the part "such as xxxx" forward to as "human intervention, such as xxxx, which are xxxx"
Our reply: We agree with your comment.

L68: suggest changing "a large river" to "the largest river in China in terms of mean discharge" to emphasize the importance of Yangtze River.
Our reply: We agree with your comment.

L92: suggest changing "that have been mainly been concerned with" to "on", making it more concise.
Our reply: We agree with your comment.

L114: change "the TGD seasonal regulation effect" to "the effect of TGD seasonal regulation"
Our reply: We agree with your comment.

L119, L121: suggest using past tense of phase, to be consistent to the phase you used at the beginning, where you used "adopted".
Our reply: We agree with your comment.

L138: change "Downstream of" to "Downstream"
Our reply: We agree with your comment.

L143: change "discharge" to "was discharged"
Our reply: We agree with your comment.

L148: change "a tidal range that extends up to 4.6m" to "a tidal range of up to 4.6m"
Our reply: We agree with your comment.

L157: Delete "Sketch", it is actually a map, not a sketch one. Suggest changing "displaying the location of gauging and hydrological stations" to "with the location of tidal gauging and hydrological stations shown with black solid circles and rec solid rectangles".
Our reply: We agree with your comment.

L168: change "difference of" to "difference between"
Our reply: We agree with your comment.

L169: "and a half" to "and dividing by two".
Our reply: We agree with your comment.

L170: "water levels of xx stations" to "water level at xx stations"
Our reply: We agree with your comment.

L206: will the solution for rectangle lateral shape channel be different with those with a V-shape? It is better to state here why such an assumption is valid as most part of estuary is not rectangle shape but v-shape.
Our reply: We agree that most small estuaries are characterized with a V-shaped cross section. However, the Yangtze estuary is extremely large with the mouth width of around 90 km, and the width of river channel is convergence from around 10 km in the downstream to around 2-3 km in the upstream. In contrast, the depth is only at around 10-20 m. In this sense we believe the rectangle lateral shape assumption is reasonable. Following the suggestion of reviewer, we shall revise the sentence as "*We further assume a nearly rectangular cross-section, considering a large width to depth ratio; hence, the tidally averaged depth is given by* $\bar{h} = \bar{A}/\bar{B}$."

L214-217: These symbols are not used for the four number and it is not appropriate to use "where" here. I suggest to move it as a note under the table 1, or express the formula for each number explicitly in the text (say, each number is described with its corresponding formula in the text).

Our reply: In the revised paper, we shall revise the sentence as: "*The definitions of these four variables are defined in Table 1, where where η is the tidal amplitude, υ is the velocity amplitude, $U_r$ is the river flow velocity, ω is the tidal frequency, $r_s$ is the storage width ratio accounting for the effect of storage area (i.e. tidal flats or salt marshes), and $c_0$ is the classical wave celerity defined as $c_0 = \sqrt{g\bar{h}/r_s}$.*"

L280: "in upstream stations" to "at upstream stations"
Our reply: We agree with your comment.

L302-309: Isn't it necessary to describe why some segments has seen little change or even decrease? It is confusing. Why increased damping denotes weaker friction? For classic understanding, it is thought larger friction lead to a higher damping rate.

Our reply: Here it should be noted that the value of damping rate $\delta_H$ is negative, thus a higher damping rate indicates less friction rather than larger friction.

Figure 5: what does each data point stand for? Monthly value? Or yearly value?
Our reply: They are monthly averaged values. In the revised paper, we shall explicitly mention this in the caption of the figure.

Section 4.3: the second part in section 4.1 is suggested to move into section 4.3.
Our reply: We do not agree this comment since the second part in section 4.1 describing the observed tidal damping rate and residual water level slope as a function of observed freshwater discharge at Datong hydrological station. This part still belongs to the observational analysis rather than the analytical analysis.

L373: "identify" seems not a good word here.
Our reply: In the revised paper, we shall replace "identify" with "observe".

L383: in "the larger the freshwater discharge is, the smaller the velocity number and the phase lag are.", suggest changing as "the larger the freshwater discharge, the smaller the velocity number and the phase lag."
Our reply: We agree with your comment.

Figure 6: why there is sharply jumping in the curve, due to different manning coefficient?

Our reply: Indeed, the discontinuous jump has to do with the adoption of two very different Manning-Strickler friction coefficient. To avoid such a sharp jump in the curve, in the revised paper, we shall adopt a friction coefficient of $K$=80-55 m$^{1/3}$s$^{-1}$

(indicating a linear reduction of the friction coefficient) over the transitional reach ($x$=32-52 km).

L416: "the river immediately downstream eroded" to "the river bed immediately downstream was eroded"
Our reply: We agree with your comment.

L426: "therefore" may be a better word than "consequently"
Our reply: We agree with your comment.

L463-466: this whole sentence reads awkward. Suggesting changing "where the tidal influence dominates that of the freshwater discharge" to "where tidal influence overwhelms the influence from freshwater discharge".
Our reply: We agree with your comment.

L500: "drawn lines" to "solid lines"
Our reply: We agree with your comment.

L519: it is not clear how you determine the value "20-yr" and "10-yr" here.
Our reply: Generally, these values should be determined based on the long-term time series of the monthly averaged high-water levels. To avoid confusing, we shall revise the sentence as: "*The corresponding flood prevention standard, therefore, is reduced due to the increased high-water level (see also Nakayama and Shankman, 2013).*"

L574: suggest changing "as a significant case study" to "as an example"
Our reply: We agree with your comment.

References
Cai, H., Savenije, H. H. G., Garel, E., Zhang, X., Guo, L., Zhang, M., Liu, F., and Yang, Q., 2018. Seasonal behaviour of tidal damping and residual water level slope in the Yangtze River estuary: identifying the critical position and river discharge for maximum tidal damping, Hydrol. Earth Syst. Sci. Discuss., doi:10.5194/hess-2018-524, in review.
Mei, X., Dai, Z., Darby, S. E., Gao, S., Wang, J., Jiang, W., 2018. Modulation of extreme flood levels by impoundment significantly offset by floodplain loss downstream of the Three Gorges Dam. Geophys. Res. Lett., 45, 3147–3155, doi:10.1002/2017GL076935
Nakayama, T., Shankman, D., 2013. Impact of the Three-Gorges Dam and water transfer project on Changjiang floods. Global Planet Change, 100: 38-50, doi:10.1016/j.gloplacha.2012.10.004.
Shaikh, B.Y., Bansal, R.K., Das, S.K., 2018. Propagation of tidal wave in coastal terrains with complex bed geometry. Environmental Processes, 5(3), 519-537.

---

## Referee Comment (RC2) · Matt Lewis (Referee) · 26 Mar 2019

A very interesting study, with applications to all "downstream consequences from land management practice (e.g. reservoirs, hydro-electric, flood risk mitigation). I think the article is great and worthy of publication, but I have some concerns – listed below. Applying an analytical model to find the downstream change to volume of a river due to upstream water collection (the three gourges dam) is neat – but I am unsure how this can be used to assess impact to biology. 1. Inter-annual variability I think some effort to resolve inter-annual variability would have been nice. Standard deviation could be added to the mean values in Figure 11 - and then a conclusion of "significant change

between months 7 to 11" can be made with confidence. At present such a statement cannot be made: Significant compared to what? Where is the test of significance? At best the authors can say "the change in the mean is clear for months 7-10". If Table 2 had more data added, i.e. how the monthly mean changes each year – it would be nice. Certainly the data is sufficient (it spans multiple years), and so the inter-annual variability can be added to Figure 11. That said, perhaps the authors can defend my comment here?

2. Sub-monthly variability impact Another concern I have is the resolution of the model. Is the frequency of boundary forcing information sufficient to resolve extreme events? For example, daily-averaged flow rates were found to be insufficient to resolve flood risk and water quality within estuary hydrodynamic models (e.g. Robins, P.E., Lewis, M.J., Freer, J., Cooper, D.M., Skinner, C.J. and Coulthard, T.J., 2018. Improving estuary models by reducing uncertainties associated with river flows. Estuarine, Coastal and Shelf Science, 207, pp.63-73.) I guess I am simply asking: you have monthly means, but how does this down-scale to hourly means, which are likely to be important for impact to wildlife and estuary impact? For this second comment, perhaps a sensitivity test is needed to prove to the reader that you can take coarse river data and resolve estuary impact. However, perhaps this can also be defended by the authors?

3. Assumptions of river geometry variability For the analytical solution method - how is river width treated for application to volume temporally variance? Is an assumption made about the river being canalised? i.e. constant bank full width? Or is there an associated flood plan? How is river depth calculated? If so, how does this effect your results?

---

## Author Comment (AC2) · 31 Mar 2019

**Responses to comments by Reviewer #2**

We thank Dr. Lewis's comments of our work. In this rebuttal, we have addressed all the comments formulated by the Reviewer by replying (in black) to his remarks (in blue).

**General comments:**

A very interesting study, with applications to all "downstream consequences from land management practice (e.g. reservoirs, hydro-electric, flood risk mitigation). I think the article is great and worthy of publication, but I have some concerns – listed below. Applying an analytical model to find the downstream change to volume of a river due to upstream water collection (the three gorges dam) is neat – but I am unsure how this can be used to assess impact to biology.

Our reply: Thanks a lot for the reviewer's positive evaluation of our manuscript. Due to the fact that in this study we mainly focus on the impacts of freshwater regulation of TGD on spatial-temporal patterns of tide-river dynamics in the Yangtze River estuary, we did not provide details concerning the TGD's impact on biology in the paper. However, we do mention the possible influence of TGD's operation on ecology in the sections of ABSTRACT and INTRODUCTION. In particular, the results obtained from this study can further be used to assess the impacts of TGD's operation on salt intrusion (as a general predictor of the aquatic ecosystem health in estuarine environment) when combined with an ecological or salt intrusion model. This is further elaborated in the DISCUSSION part (see Section 5.4 in the manuscript). In the revised paper, we shall explicitly mention that "*However, to quantify the potential impacts of TGD's operation on salt intrusion and related aquatic ecosystem health in general, it is required to couple the hydrodynamic model to the ecological or salt intrusion model (e.g., Qiu and Zhu, 2103; Cai et al., 2015).*"

**Major comments**

1. Inter-annual variability. I think some effort to resolve inter-annual variability would have been nice. Standard deviation could be added to the mean values in Figure 11 - and then a conclusion of "significant change between months 7 to 11" can be made with confidence. At present such a statement cannot be made: Significant compared to what? Where is the test of significance? At best the authors can say "the change in the mean is clear for months 7-10". If Table 2 had more data added, i.e. how the monthly mean changes each year – it would be nice. Certainly the data is sufficient (it spans multiple years), and so the inter-annual variability can be added to Figure 11. That said, perhaps the authors can defend my comment here?

Our reply: We thank the reviewer for this comment. Indeed, it is better to resolve the inter-annual variability. In the revised paper, we shall include the standard deviation information in Figure 11 (see Figure R1 below).

[Figure]

Figure R1. Temporal variation of the position of the tidal limit relative to the TSG station for both the pre-TGD and the post-TGD periods. The vertical error bar at each data point indicates the standard deviation of the analytically computed time series.

2. Sub-monthly variability impact. Another concern I have is the resolution of the model. Is the frequency of boundary forcing information sufficient to resolve extreme events? For example, daily-averaged flow rates were found to be insufficient to resolve flood risk and water quality within estuary hydrodynamic models (e.g. Robins, P.E., Lewis, M.J., Freer, J., Cooper, D.M., Skinner, C.J. and Coulthard, T.J., 2018. Improving estuary models by reducing uncertainties associated with river flows. Estuarine, Coastal and Shelf Science, 207, pp.63-73.) I guess I am simply asking: you have monthly means, but how does this down-scale to hourly means, which are likely to be important for impact to wildlife and estuary impact? For this second comment, perhaps a sensitivity test is needed to prove to the reader that you can take coarse river data and resolve estuary impact. However, perhaps this can also be defended by the authors?

Our reply: Due to the fact that the main purpose of this study lies in quantifying the impacts of TGD's seasonal regulation on the tide-river dynamics over the entire reach of the Yangtze River estuary, thus we adopted the monthly averaged river discharge conditions. This is possible to down-scale to the tidally averaged means since the proposed analytical model is obtained based on the tidally averaged conditions. For such a kind of application using tidally averaged means, the reviewer can kindly refer to our previous publications of Cai et al. (2014, 2016). However, the model cannot be used to understand the impacts of hourly varying freshwater discharge on the tide-river dynamics because of model limitation. To resolve extreme events and their

impacts on flood control and water quality, as suggested by the reviewer (e.g., Robins et al., 2018), it is required to use a high-resolution numerical model adopting high-resolution boundary conditions (e.g., hourly mean river discharge).

3. Assumptions of river geometry variability. For the analytical solution method – how is river width treated for application to volume temporally variance? Is an assumption made about the river being canalised? i.e. constant bank full width? Or is there an associated flood plan? How is river depth calculated? If so, how does this effect your results?

Our reply: Indeed, in the analytical model we simplified the channel geometry to be in the shape of rectangular geometry. This means that the channel width is assumed to be time-invariant, while the water depth is variable as a function of tidal and riverine forcing. Such an assumption is particularly reasonable since the Yangtze River estuary is extremely large with the mouth width of around 90 km, and the width of river channel is convergent from around 10 km in the downstream section to around 2-3 km in the upstream section. On the other hand, the depth is only at around 10-20 m along the main course of the estuary. Consequently, the width to depth ratio is large so that the cross-sectional area variability can be primarily caused by the depth variability. The possible influence of storage area (i.e. flood plain and tidal flats) is taken into consideration by introducing the parameter of the storage width ratio $r_s$ (i.e., the ratio of the storage width to the averaged stream width). Such a kind of rectangular shape assumption has been used in many previous studies (e.g., Van Rijn, 2011, Toffolon and Savenije, 2011, Cai et al., 2014, 2016). In the revised paper, we shall clarify such an assumption: "*We further assume a nearly rectangular cross-section, considering a large width to depth ratio; hence, the tidally averaged depth is given by* $\bar{h} = \bar{A}/\bar{B}$ *and the cross-sectional area variability can be primarily due to the change in depth.*"

References:
Cai, H., Savenije, H. H. G., and Toffolon, M.: Linking the river to the estuary, influence of river discharge on tidal damping, Hydrol. Earth Syst. Sci., 18(1), 287-304, https://doi.org/10.5194/hess-18-287-2014, 2014.
Cai, H., Savenije, H.H.G., Zuo, S., Jiang, C., and Chua, V.: A predictive model for salt intrusion in estuaries applied to the Yangtze estuary, J. Hydrol., 529, 1336-1349, https://doi.org/10.1016/j.jhydrol.2015.08.050, 2015.
Cai, H., Savenije, H. H. G., Jiang, C. Zhao L., Yang Q.: Analytical approach for determining the mean water level profile in an estuary with substantial fresh water discharge, Hydrol. Earth Syst. Sci., 20, 1-19, https://doi.org/10.5194/hess-20-1-2016, 2016.
Qiu, C. and Zhu., J.: Influence of seasonal runoff regulation by the Three Gorges Reservoir on saltwater intrusion in the Changjiang River Estuary, Cont. Shelf Res., 71, 16-26, https://doi.org/10.1016/j.csr.2013.09.024, 2013.
Robins, P.E., Lewis, M.J., Freer, J., Cooper, D.M., Skinner, C.J., Coulthard, T.J.: Improving estuary models by reducing uncertainties associated with river flows,

Estuar. Coast. Shelf S., 207, 63-73, https://doi.org/10.1016/j.ecss.2018.02.015, 2018.

Toffolon, M., Savenije, H. H. G.: Revisiting linearized one-dimensional tidal propagation, J. Geophys. Res. 116, C07007, https://doi.org/10.1029/2010JC006616, 2011.

van Rijn, L. C.: Analytical and numerical analysis of tides and salinities in estuaries; part I: Tidal wave propagation in convergent estuaries, Ocean Dynam. 61(11), 1719–1741, https://doi.org/10.1007/s10236-011-0453-0, 2011.

---

## Author Response (AR2)

**Response letter**

We thank Editor's careful consideration of our work. In this rebuttal, we have addressed all the minor concerns raised by the Editor by replying (in black) to your remarks (in blue). The lines numbers in this rebuttal refer to the revised version of the manuscript.

**There are a few minor revisions**

p11 Need to define rs ( storage ratio)

**Our reply:** Thank you for pointing this out. In the revised paper, we have explicitly defined the adopted storage width ratio $r_S$ as: "*$r_S = B_S/\bar{B}$ is the storage width ratio between the storage width $B_S$ and the stream width $\bar{B}$ that accounts for the effect of storage area (i.e., tidal flats or salt marshes).*" See lines 216-218 of the revised manuscript.

p14 "This indicates a consistent enhancement of the tidal dynamics...except near ZJ station". This exception needs to be discussed in the paper.

**Our reply:** In the revised paper, we have supplemented the discussion of this point by including the following sentences: *"The exceptional case in ZJ station is likely due to the fact that ZJ station is located near the position of the tidal current limit during the dry season (Guo et al., 2015; Zhang et al., 2018). The shallow and narrow geometry around ZJ station impedes the tidal wave propagation when river discharge increases due to the TGD operation during the dry season (Chen et al., 2012), leading to a remarkably decreasing tidal range in the first half of the year."* See lines 277-283 of the revised manuscript.

p23 Figure 6. The discontinuity in the tidal damping number ( delta) and simulated velocity number in Figure 7needs to be explained.
In a numerical model this behaviour is not acceptable.

**Our reply:** We agree with your comment! In the revise paper, we have explicitly mentioned that "*Here, we used two values for K: K = 80 $m^{1/3}{\cdot}s^{-1}$ in the tide-dominated region (x = 0–32 km), and a smaller value of K = 55 $m^{1/3}{\cdot}s^{-1}$ in the river-dominated region (x = 52–450 km). In addition, to avoid sharp jump in the analytically computed parameters due to the adoption of different friction coefficients, we adopted a friction coefficient of K=80-55 $m^{1/3}s^{-1}$ (indicating a linear reduction of the friction coefficient) over the transitional reach (x=32-52 km).*" See lines 350-355 of the revised manuscript.

Meanwhile, in section 4.3 of the revised paper, we also explicitly mentioned that "*
[revised manuscript text omitted]
\left(2\alpha\right) + \frac{1}{24}\sin\left(6\alpha\right) - \frac{1}{60}\sin\left(8\alpha\right), \tag{18}$$

$$p_1 = \frac{7}{6}\sin\left(\alpha\right) - \frac{7}{30}\sin\left(3\alpha\right) - \frac{7}{30}\sin\left(5\alpha\right) + \frac{1}{10}\sin\left(7\alpha\right), \tag{19}$$

$$p_2 = \pi - 2\alpha + \frac{1}{3}\sin\left(2\alpha\right) + \frac{19}{30}\sin\left(4\alpha\right) - \frac{1}{5}\sin\left(6\alpha\right), \tag{20}$$

$$p_3 = \frac{4}{3}\sin\left(\alpha\right) - \frac{2}{3}\sin\left(3\alpha\right) + \frac{2}{15}\sin\left(5\alpha\right). \tag{21}$$

The coefficients $p_1$, $p_2$, and $p_3$ determine the magnitudes of the linear, quadratic, and cubic frictional interaction, respectively.